# Mechanisms of maternal antibody interference with rotavirus vaccination

Tawny L Chandler[1,5], Sarah Woodyear [ID][1,5], Valerie Chen[1], Tom M Lonergan[1], Natalie Baker [ID][1], Katherine Harcourt[2], Simon Clare[2], Faraz Ahmed [ID][3] & Sarah L Caddy [ID][1,4 ✉]

## Abstract

**Maternal antibodies are transferred transplacentally to fetuses and then through lactation to infants to protect them whilst their own immune system is still immature. However, these maternal antibodies also suppress neonatal B-cell responses, thereby impairing vaccine efficacy and leaving infants potentially vulnerable to life-threatening pathogens, such as rotaviruses. Currently available rotavirus vaccines are composed of live-attenuated viral strains administered to infants orally at 6-8 weeks old. Although high concentrations of maternal antibodies correlate with poor production of antibodies following vaccination (i.e., seroconversion), the immunological basis of this interference is unknown. To investigate the underlying mechanisms, we here developed a mouse model of neonatal oral rotavirus vaccination, in which vaccination only fails to induce seroconversion if maternal antibodies are present. Such antibodies are shown to block vaccine replication, while faster maternal antibody waning is observed in vaccinated compared to unvaccinated pups. FcγRIIB deletion does not overcome interference in pups, although pup IgG levels increase when maternal antibody titers are very low. Our findings show that maternal antibody-mediated vaccine clearance is a key mechanism of interference with oral rotavirus vaccines, with a minor role for FcγRIIB in neonatal IgG responses.**

**Keywords** Maternal Antibody; FcγRIIB; Rotavirus; Vaccine
**Subject Categories** Immunology; Microbiology, Virology & Host Pathogen Interaction

See also: A Mollin & SN Langel

## Introduction

Neonates of all mammalian species are considered immunologically naïve at birth. This places the neonate at a high risk of infection while their immune system is maturing. To counter this, mothers have evolved a strategy to deliver immune protection to their offspring in the form of maternal antibodies (MatAbs). Antibodies in the maternal circulation can be passively transferred to the fetus across the placenta, and postnatally to the neonate in milk. These MatAbs function to protect the naïve offspring against pathogens faced in the environment. Lower levels of MatAbs can be correlated with increased infectious disease in the infant (Niewiesk, 2014).

Whilst MatAbs are an undoubted evolutionary advantage, a paradoxical effect of their presence is a reduction in vaccine efficacy in infants. Often known as MatAb interference or MatAb blunting, the phenomenon of lower vaccine efficacy in the face of MatAbs has been described for many different vaccines. This includes numerous viruses of human and veterinary importance, including measles virus (Albrecht et al, 1977), foot and mouth disease virus (Kitching and Salt, 1995), hepatitis A virus (Kanra et al, 2000), and poliovirus (Perkins et al, 1958). MatAb interference is especially concerning for rotavirus vaccines. Rotavirus is a significant cause of gastroenteritis in young children, attributed to over 200,000 deaths in children under five years old (Cohen et al, 2022). The current rotavirus vaccines are live-attenuated strains that were first licensed in 2006, and although these vaccines have proven to be highly effective in high-income countries, their efficacy is often less than 50% in low- and middle-income regions. Multiple factors have been proposed to be responsible for this poor efficacy, including malnutrition, co-infections, and host genetics, but a leading explanation is MatAb interference (Desselberger, 2018; Otero et al, 2020; Clarke and Desselberger, 2015). MatAb titers are generally higher in lower-middle-income countries, which could account for this geographical variation in vaccine efficacy (Rimer et al, 1992; Moon et al, 2010; Novak and Svennerholm, 2015). A substantial number of clinical studies have identified a negative association between high titers of MatAbs and poor seroconversion following vaccination with currently used rotavirus vaccines (Moon et al, 2016; Chilengi et al, 2016; Becker-Dreps et al, 2015; Appaiahgari et al, 2014; Lee et al, 2018; Parker et al, 2021).

The mechanisms by which MatAbs can reduce vaccine efficacy have been of significant interest over many years, yet remain unclear. A number of different theories have been proposed, including MatAb-mediated masking of vaccine epitopes, and direct inhibition of neonatal B cells via MatAbs cross-linking the B cell receptor (BCR) and the inhibitory FcγRIIB receptor (Siegrist, 2003;

[1]Baker Institute for Animal Health, Cornell University, Ithaca, NY 14850, USA. [2]Department of Medicine, University of Cambridge, Cambridge CB2 0QH, UK. [3]Genomics Innovation Hub and TREx Facility, Institute of Biotechnology, Cornell University, Ithaca, NY 14850, USA. [4]Department of Microbiology and Immunology, College of Veterinary Medicine, Cornell University, Ithaca, NY 14850, USA. [5]These authors contributed equally: Tawny L Chandler, Sarah Woodyear. ✉E-mail: sarahcaddy@cornell.edu

Niewiesk, 2014). Clearance of vaccines by MatAbs so the neonatal immune response remains naïve has also been a leading theory, but some studies have shown that B cell activation still occurs in the presence of high levels of MatAbs (Vono et al, 2019). Ultimately, the diversity of immune responses needed to confer protection against distinct pathogens following vaccination may limit our ability to apply conclusions from one experimental vaccine model to another; however, each experimental vaccine model still offers unique insights and the ability to reveal diverse immune mechanisms.

In this study, we aimed to understand the role of MatAbs in the context of rotavirus vaccination, given the widely reported issues with rotavirus vaccine efficacy in human infants. Prior research in the field of MatAb interference mechanisms has largely focused on systemic or respiratory pathogens (Kim et al, 2011; Vono et al, 2019), and we predicted that responses to a gastrointestinal pathogen and orally delivered vaccine would be distinct. Here, we successfully established a mouse model of MatAb interference to a live-attenuated rotavirus vaccine, and used this to investigate the potential mechanisms responsible for poor vaccine efficacy in the presence of MatAbs. Although there are some biological differences in MatAb transfer between mice and humans, for example, IgG is transferred both transplacentally and via milk in mice, whereas in humans it occurs predominantly through the placenta, important similarities remain. The tractability of mouse models, along with the presence of a hemochorial placenta in both species, makes the mouse a valuable system for studying maternal antibody transfer. Using a combination of hypothesis-driven experimental approaches, we have shown that the major mechanism of interference is MatAb-driven clearance of the rotavirus vaccine prior to vaccine encounter by the neonatal immune system.

# Results

## Rotavirus-specific maternal antibodies block seroconversion to rotavirus vaccination in pups

Rotavirus-specific MatAbs in humans have been associated with reduced infant seroconversion to oral rotavirus vaccines in multiple clinical studies (Chilengi et al, 2016; Moon et al, 2016; Becker-Dreps et al, 2015; Lee et al, 2018; Parker et al, 2021; Appaiahgari et al, 2014). To study the immunological mechanisms underpinning this observation, we aimed to establish a mouse model of rotavirus-specific MatAb transfer and vaccination. We began by infecting adult female C57BL/6 mice with 10 FFU of an attenuated murine rotavirus strain EMcN known to readily infect this mouse line (McNeal et al, 2004). Infection was performed by oral gavage, with control mice receiving an equal volume of PBS. Female mice were mated with male BALB/c mice after seroconversion was confirmed by ELISA at 7–10 days post infection. C57BL/6 x BALB/c pups were vaccinated at 7 days old using a subclinical dose of EMcN rotavirus delivered by oral gavage. This pup vaccine timepoint was selected for all experiments unless stated, as mouse pups are readily susceptible to rotavirus when 7 days old, and also capable of mounting a rotavirus-specific IgG response at this age (Du et al, 2017; Wolf et al, 1981). This timepoint also approximately models the age at which human infants can mount an IgG response to rotavirus vaccination (6–8 weeks of age)

(Armah et al, 2016). All pups within a litter received the same treatment, and both sexes were considered in analyses. Pups were weaned at 3 weeks old, and blood samples were collected every 2 weeks from 4–10 weeks of age. The experimental pipeline is presented in Fig. 1A.

Serum antibodies circulating in the pups post weaning were quantified by a rotavirus-specific sandwich ELISA. As shown in Fig. 1B, rotavirus-specific IgG from the dam was readily detectable in a litter of pups born to a seropositive dam ("MatAbs only") at 4 weeks of age. A single dilution of sera (1:200) was studied due to the limited sample volume at this early timepoint, and for comparison, longitudinally, all future timepoints were analyzed at the same dilution. MatAbs were observed to wane over the subsequent 6-week period. A second litter of pups that were vaccinated with live-attenuated murine rotavirus in the absence of any MatAbs ("Vaccine only") made a robust rotavirus-specific IgG response. However, a third litter of pups that received murine rotavirus vaccine in the presence of MatAbs ("Vaccine + MatAbs") failed to mount their own detectable serum IgG response. This showed that in our experimental model, MatAbs negatively affected the ability of pups to seroconvert post vaccination, in agreement with observations made in human studies.

One advantage of using a mouse model over human samples is the ability to differentiate MatAbs detected in pup serum from those made by the pup themselves. We used two alternative methods to verify that pups vaccinated in the presence of MatAbs did not produce a detectable antibody response. Firstly, our use of C57BL/6 dams and BALB/c males took advantage of variation in IgG1 alleles within these two mouse strains. BALB/c mice have the IgG1a allele, whereas C57BL/6 mice have IgG1b. Use of an IgG1a-specific secondary antibody to quantify serum antibodies by ELISA only detected IgG1 produced by the pup. As shown in Fig. 1C, when pups were 10 weeks old, it was clear that only pups receiving the oral vaccine in the absence of MatAbs mounted their own IgG1 response. As rotavirus is a mucosal pathogen, an alternative means of differentiating maternal and neonatal antibody responses is by quantification of IgA. IgG from the mother's milk can enter the pup's circulation due to FcRn-mediated uptake across the intestinal barrier (Israel et al, 1995), but IgA does not cross the intestinal epithelium; thus, any IgA detected in pup serum must be produced by the pup themselves. Figure 1D shows serum IgA of 10-week-old mice born to mothers with or without antibodies, and mirrors Fig. 1C accordingly, in addition to confirming that no maternal IgA enters the pup circulation. We also wanted to verify that the production of IgA detected in stool was similarly impacted by the presence of MatAbs. To achieve this, we independently vaccinated a second cohort of litters with and without MatAbs, and longitudinally collected stool samples. IgA ELISA results presented in Fig. 1E again show that no stool IgA is detected in the presence of MatAbs. Overall, these results support the conclusion that MatAbs interfere with the ability of the pup to seroconvert to rotavirus vaccination.

Many strains of human rotavirus circulate in human populations, yet rotavirus vaccines are currently restricted to either a single strain (G1P[8] e.g., Rotarix and GSK), or five reassortant strains (e.g., Rotateq and Merck). This means that MatAbs transferred are likely to target rotavirus strains that differ from those in the vaccine administered to the neonate. To address whether MatAb interference can still occur in our model when the

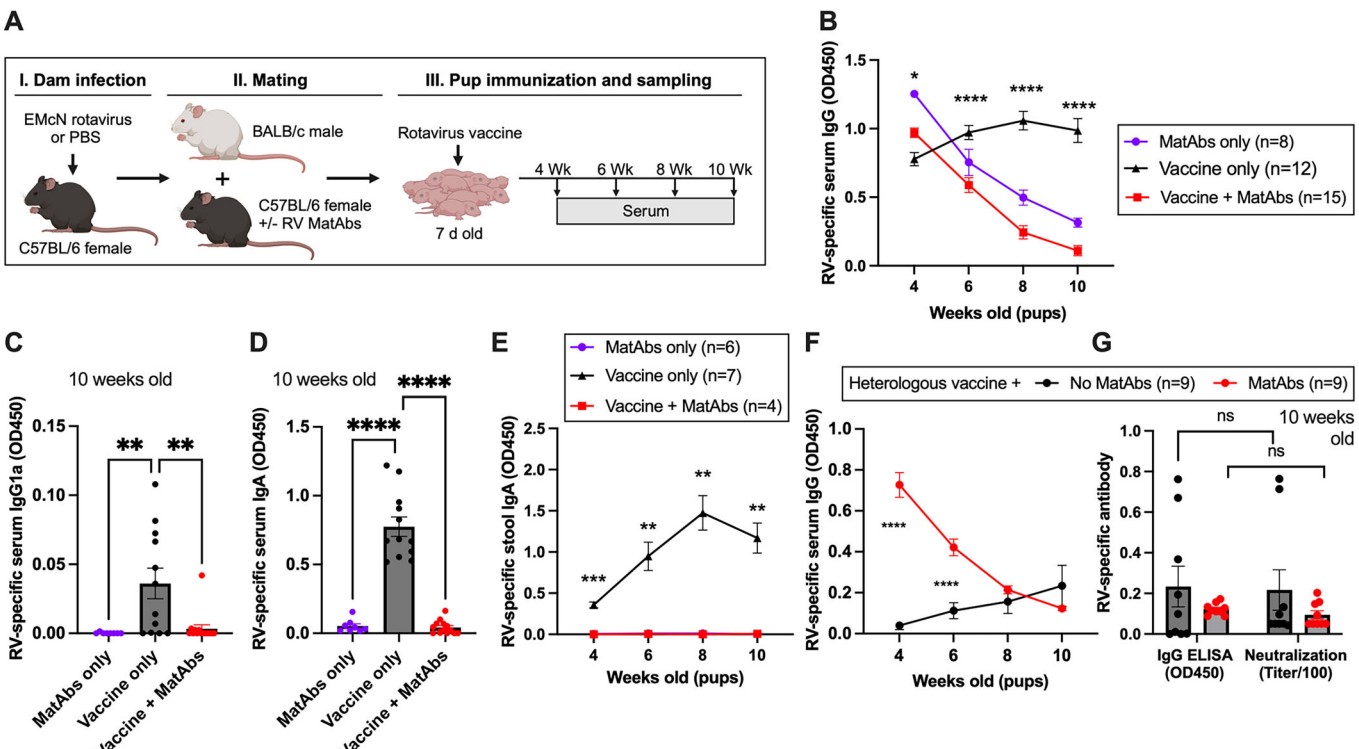

**Figure 1. Maternal antibodies interfere with pup immune responses.**

(A) Schematic diagram of experimental breeding and infection protocol using C57BL/6 female and BALBc male mice. (B) Serum rotavirus-specific IgG in pups as measured by ELISA from 4 to 10 weeks of age; $p = 0.01$ at 4 wk and $p < 0.0001$ at 6, 8, and 10 wk for presence vs absence of MatAbs at vaccination. (C, D) Differentiation of maternal and pup antibodies by (C) IgG1a and (D) IgA at 10 weeks of age; $p = 0.006$, and $p = 0.003$ for MatAbs only vs vaccine only, and vaccine only vs vaccine + MatAbs, respectively (C) and $p < 0.0001$ for MatAbs only vs vaccine only, and vaccine only vs vaccine + MatAbs in (D); $n = 8$, $n = 12$, and $n = 15$ for MatAbs only, vaccine only, and vaccine + MatAbs, respectively. (E) Stool IgA titers in the presence or absence of MatAbs; $p < 0.0001$, $p = 0.004$, $p = 0.001$, and $p = 0.002$ at 4, 6, 8, and 10 wk, respectively, for presence vs absence of MatAbs at vaccination. (F) Longitudinal analysis of serum IgG by ELISA from pups vaccinated with a heterologous rotavirus vaccine strain; $p < 0.0001$ and $p = 0.002$ at 4 and 6 wk, respectively. (G) Comparison of IgG quantified by ELISA and serum neutralization titers in 10-week-old mice from (F). Data information: Statistical significance was determined by two-way ANOVA with repeated measures (B, E, F), one-way ANOVA (C, D), or paired t-test (G). Significant Tukey's adjusted pair-wise comparisons of vaccination in the absence or presence of MatAbs (B, E) and Bonferroni-corrected pair-wise comparisons (C, D, F) are shown; error bars indicate SEM. Source data are available online for this figure.

vaccinating strain and MatAb specificity are mismatched, we tested a heterologous vaccine approach. Dams were seropositive to the murine rotavirus strain (G16P[16]) as before, whereas pups were vaccinated with a G1P[8] human rotavirus strain. We have previously shown that murine rotavirus-specific antibodies can neutralize human rotavirus strains, albeit to a lesser degree than for homologous strains (Woodyear et al, 2024). No pups showed evidence of seroconversion to the human rotavirus vaccine in the presence of MatAbs, whereas some degree of seroconversion was apparent in the naïve pups. This is shown in the longitudinal results in Fig. 1F, with a focused view of week 10 samples presented in Fig. 1G. The latter figure also presents the results of serum neutralization assays with the human rotavirus vaccine strain, demonstrating a high degree of agreement between the binding and functional assays. It is important to note that the limited seroconversion identified in the no MatAb group is likely due to the species-specificity of rotaviruses (Woodyear et al, 2024; Sánchez-Tacuba et al, 2022). Further experiments in this study, therefore, continued to focus on murine-specific rotaviruses to maximize potential seroconversion.

## Germinal center formation is limited in the presence of maternal antibodies

Whilst no pup antibody was detected in the serum of pups vaccinated in the presence of MatAbs, we questioned what effect MatAbs were having on the wider neonatal B cell response. To address this, we repeated our experimental timeline with administration of rotavirus or PBS to dams, transfer of any induced MatAbs to pups, and then vaccination of pups at 7 days of age. Pups were culled 20 days post vaccination, and mesenteric lymph nodes (MLN) were collected from all mice for analysis by flow cytometry or histology.

We observed a significant decrease in the formation of germinal centers in MLN of mice vaccinated in the presence of MatAbs. This was apparent via flow cytometry, where a statistically significant difference was detected (Fig. 2A, germinal centers identified by B220, GL7, and Fas staining). We verified this result using both haematoxylin and eosin (H&E) staining (Fig. 2B) and immuno-fluorescence (staining for Ki-67 and B220, Fig. 2C) of MLN tissue sections from pups in the same litters. Minimal visible germinal

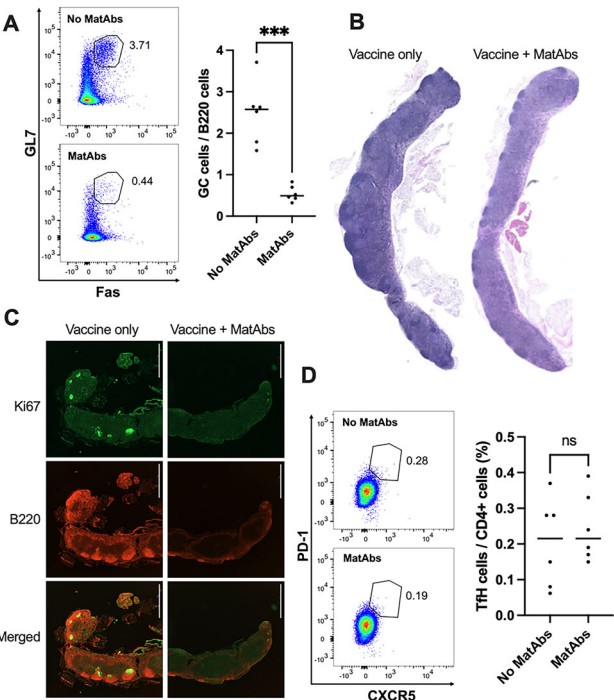

**Figure 2. Maternal antibodies limit the formation of germinal centers (GCs) in draining lymph nodes.**

(A) Flow cytometry of GCs in mesenteric lymph nodes (MLNs) from pups with ($n = 6$) or without ($n = 6$) MatAbs at 20 days post vaccination, with representative flow cytometry plots and scatterplot presenting all mice, $p = 0.0001$. (B, C) H&E staining (B) and immunofluorescence (C) of MLN sections from mice vaccinated in the presence/absence of MatAbs. Scale bars: 1000 μm. (D) Flow cytometry of T follicular helper (TfH) cells in MLNs from pups with ($n = 6$) or without ($n = 6$) MatAbs at 20 days post vaccination. The gating strategy of a representative mouse used to identify TfH cells is presented in Fig. EV1. Data information: Statistical significance was determined by unpaired two-tailed $t$-tests (***$p = 0.0001$). Source data are available online for this figure.

center formation was evident in pups vaccinated in the presence of MatAbs. This is in agreement with the observed serological responses, but in contrast to findings with a MatAb model and parenteral influenza subunit vaccination (Vono et al, 2019). Interestingly, there was no difference in the relative level of T follicular helper (TfH) cells in MLNs (Fig. 2D). Gating strategies for GC and TfH cells are outlined in Fig. EV1.

## Maternal antibody-mediated interference is robust with very low maternal antibody doses

Our first series of experiments demonstrated complete ablation of pup seroconversion to rotavirus vaccination in the presence of MatAbs induced by infection of dams with high doses of rotavirus. As MatAb titer has been correlated with seroconversion to rotavirus vaccination in human infants (Appaiahgari et al, 2014; Chilengi et al, 2016; Becker-Dreps et al, 2015; Moon et al, 2016; Lee et al, 2018), we next asked whether pup seroconversion could be achieved by lowering the dose of MatAbs received by pups. We predicted that reducing the titer of MatAbs in the dam would enable more pups to seroconvert. This would facilitate the

comparison of antibody repertoires between the dam and the pups to examine the proposed phenomenon of MatAb-mediated epitope masking.

To lower the dose of MatAbs received by the pups, we used two alternative approaches. First, we infected the dams with two different doses of rotavirus (1 FFU and 0.1 FFU) one week prior to mating, with the expectation that maternal rotavirus-specific antibody titers would be different during pregnancy and lactation. Figure 3A presents maternal serum IgG and IgA quantified by ELISA in two dams infected with rotavirus, showing how circulating MatAbs changed following the birth of pups. Whilst serum IgA was clearly distinct between the two dams, inducing different levels of serum IgG was more challenging, and vaccination of pups appears to have a boosting effect on serum antibody levels in the dam. Analysis of serum rotavirus-specific IgG and IgA levels in pups born to the dams depicted in Fig. 3A are shown in Fig. 3B,C. We found that regardless of dam serum antibody levels, neither litter of pups with MatAbs seroconverted following rotavirus vaccination. However, as we observed that maternal IgG titers converged and likely reached a ceiling, separating MatAb delivery titer by this approach was shown to be an imperfect strategy.

For our second approach, we used a passive transfer model, whereby a controlled amount of MatAb was delivered to dams. This allowed us to titrate down how much MatAb was transferred to pups. To achieve this, serum from rotavirus-specific antibody-positive mice was pooled and administered intraperitoneally (IP) to dams 24 h prior to vaccination of pups at 7 days of age and a second time 1-week post vaccination. A key difference using this experimental approach as opposed to naturally induced MatAbs was that MatAb delivery via passive transfer was only via milk to nursing pups. Verification of rotavirus-specific antibody transfer from the intraperitoneal space to the maternal circulation and milk is shown in Fig. 3D. Antibody titers in naturally infected dams are included for comparison. A log10 scale was used for the x-axis to demonstrate that IgA was detectable in the milk of the dam receiving 50 μl serum, albeit at a very low level. IgG and IgA titers in the serum pool used for passive transfer are shown in Fig. EV2C. Unexpectedly, as shown in Fig. 3E, no seroconversion was detectable in either litter of pups receiving MatAbs from passively transferred dams. As rotavirus-specific serum IgA responses were low in all pups (Fig. EV2A), rotavirus-specific IgA was also quantified in stool pellets collected from individual mice at each timepoint. Again, no seroconversion was demonstrated if pooled serum was administered to dams (Fig. 3F). Overall, this shows that even very low titers of MatAbs delivered only by milk can interfere with infant vaccine responses in our model.

## FcγRIIB signaling is not a major mediator of maternal antibody interference

One of the major mechanisms proposed for MatAb interference has been premature inhibitory B cell signaling by IgG MatAbs binding to FcγRIIB (Niewiesk, 2014; Edwards, 2015; Siegrist, 2003). It has been theorized that IgG MatAbs can function to co-ligate both the BCR and FcγRIIB on infant B cells simultaneously, resulting in blockade of B cell activation. To test this hypothesis, we replicated our mouse model of maternal interference to rotavirus vaccination in FcγRIIB knockout (KO) mice (kindly gifted from Jeffrey

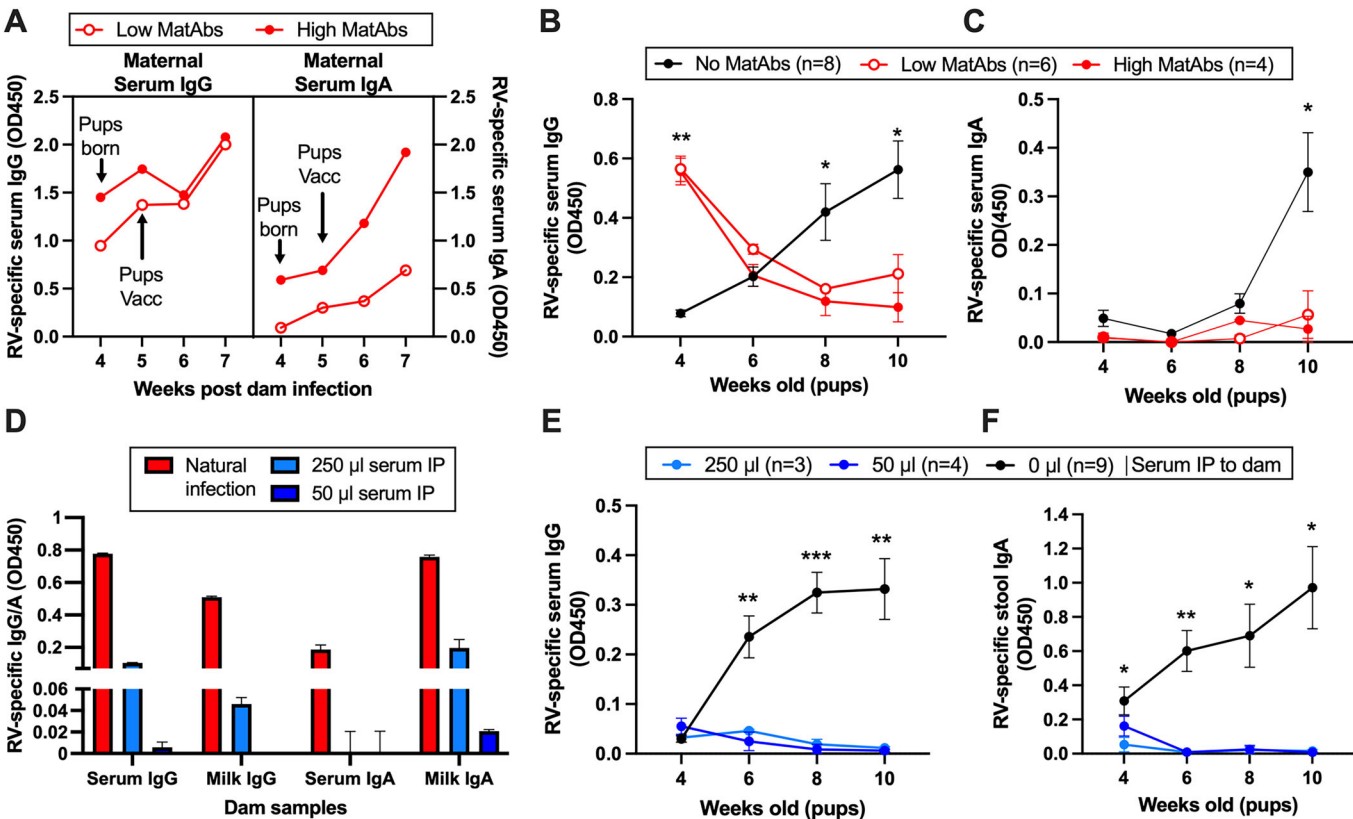

**Figure 3. Maternal antibody-mediated interference is robust with low antibody doses.**

(A) Longitudinal measurement of dam rotavirus-specific antibody by ELISA for a low MatAbs ($n = 1$) and a high MatAbs ($n = 1$) mouse. (B, C) Longitudinal measurement of serum rotavirus-specific IgG (B) and IgA (C) in pups receiving high, low, or no MatAbs; $p = 0.0025$, $p = 0.05$, $p = 0.005$ at 4, 8, and 10 wk, respectively, for no MatAb vs high MatAbs and $p = 0.0001$, $p = 0.03$ at 4 and 10 wk, respectively, for no MatAb vs low MatAb in (B), and $p = 0.03$, $p = 0.01$ for low or high MatAb, respectively, vs no MatAb. (D) Quantification of rotavirus-specific IgG and IgA at the time of pup vaccination in the serum and milk of dams following natural infection ($n = 1$) or receiving 50 µL ($n = 1$) or 250 µL ($n = 1$) of pooled serum from infected mice intraperitoneally (IP). (E, F) Longitudinal measurement of rotavirus-specific serum IgG (E) and stool IgA (F) in pups receiving high, low or no MatAbs; $p = 0.005$, $p = 0.0003$, $p = 0.005$ at 6, 8, and 10 wk, respectively, for 250 vs 0 µL serum and $p = 0.003$, $p = 0.0002$, $p = 0.0004$ at 6, 8, and 10 wk, respectively, for 50 vs 0 µL serum in (E); $p = 0.05$, $p = 0.003$, $p = 0.02$, $p = 0.02$ at 4, 6, 8, and 10 wk, respectively for 250 vs 0 µL serum and $p = 0.003$, $p = 0.02$, $p = 0.02$ at 6, 8, and 10 wk, respectively for 50 vs 0 µL serum in (F). Data information: Error bars show standard error of the mean, representing biological replicates (B, C, E, F) or technical replicates of a single mouse (A, D). Statistical significance was determined by two-way ANOVA with repeated measures (B, C, E, F). The largest significant Tukey-adjusted pair-wise comparisons between the absence or presence of MatAbs (B, C) and the absence of MatAb or passively transferred serum are shown (E, F) (*$p \leq 0.05$; **$p < 0.01$; ***$p < 0.001$; ****$p < 0.0001$). Source data are available online for this figure.

Ravetch, Rockefeller). FcγRIIB KO mice have been reported to have higher antibody titers due to the absence of signaling to bring an initial B cell response to a close (Takai et al, 1996), we therefore sought to evaluate any potential differences in rotavirus-specific IgG titers following primary infection. As shown in Fig. 4A, both wildtype (WT) and KO mice exhibited a steady increase in IgG titers over the 5-week period post infection. A comparable titer of MatAb was therefore expected to be delivered to pups, ensuring MatAb titer was not a variable impacting the results.

Figure 4B shows the seroconversion of KO pups vaccinated with rotavirus in the presence or absence of MatAbs. MatAb interference is clear in the group of pups receiving MatAbs from naturally infected dams, with pup serum IgG titers only increasing in the absence of MatAbs. This indicates that FcγRIIB signaling is not essential for MatAb interference in this experimental set up. We next asked whether FcγRIIB could be playing a more important role when MatAb titer was very low. As we had already established that the most reproducible approach to ensuring low titers of

MatAbs was passive transfer of pooled serum from seropositive mice, we repeated this experimental strategy in FcγRIIB KO mice. We selected the same low dose of polyclonal immune sera (50 µl) that induced interference in WT mice (Fig. 3D–F), and contained both IgG and IgA. This was administered to two lactating dams 24 h before pups were vaccinated at 7 days old. Two control dams did not receive any treatment. The fours litters of pups were then longitudinally sampled post weaning for analysis of their antibody responses. As shown in Fig. 4C, MatAb interference with serum IgG responses was no longer observed when MatAb titers were low, in direct contrast to Fig. 4B with high MatAb titers. This indicates that FcγRIIB can drive IgG-mediated MatAb interference when very low doses of MatAbs are delivered via milk. However, when pup serum and stool IgA responses were quantified by ELISA, there was no evidence of pup IgA production in the presence of MatAbs at any timepoint. This demonstrates that very low doses of MatAbs still mediate interference with systemic pup IgA responses in the absence of FcγRIIB signaling. This is the first indication that

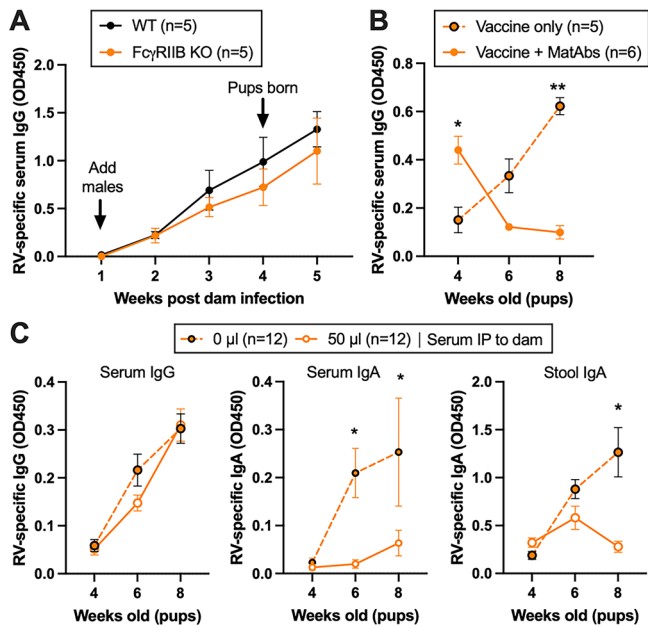

**Figure 4. Impact of FcγRIIB signaling on maternal antibody interference.**

(A) Serum IgG responses of wildtype (WT) and FcγRIIB knockout (KO) dams following rotavirus infection as shown by ELISA. $n = 5$ per group. (B) Evaluation of seroconversion by KO pups to rotavirus vaccination in the presence/absence of maternal antibodies by serum IgG ELISA; $p = 0.02$ and $p < 0.0001$ at 4 and 8 wk, respectively. (C) Serum IgG, serum IgA and stool IgA responses to rotavirus vaccination in pups vaccinated 24 h after dams were administered rotavirus-positive serum intraperitoneally (IP); $p = 0.03$ and $p = 0.03$ at 6 and 8 wk, respectively for serum IgA and $p = 0.003$ at 8 wk for stool IgA. Data information: Statistical significance was determined by two-way ANOVA with repeated measures. Significant Tukey's adjusted pair-wise comparisons between vaccination in the absence or presence of MatAbs or passively transferred serum are shown (*$p \leq 0.05$ and **$p < 0.0001$); error bars indicate SEM. Source data are available online for this figure.

interference with pup IgG and IgA responses can occur via separate mechanisms.

## Maternal antibodies mediate vaccine clearance

A leading hypothesis for the mechanism by which MatAbs interfere with vaccination response in neonates is clearance or neutralization of the vaccine particles (Niewiesk, 2014; Edwards, 2015; Siegrist, 2003). To investigate whether rotavirus-specific MatAbs altered replication of the murine live-attenuated vaccine, we sought to quantify vaccine replication in the pup gastrointestinal tract. Two litters of 7-day-old mice were vaccinated in the presence of MatAbs, and two litters from naïve dams were vaccinated. Stool samples were collected once daily from every pup for 7 days after vaccination. Samples from each litter were pooled to generate sufficient material for analysis. qPCR was performed on nucleic acid extracted from stool samples and showed that replication of the viral vaccine was substantially reduced in the presence of MatAbs (Fig. 5A). This suggests MatAbs in the intestinal mucosa are binding to viral vaccine particles and blocking replication.

In addition to reduced vaccine replication in the presence of MatAbs, we also observed a surprising reduction in rotavirus-specific IgG MatAbs in the serum of vaccinated pups compared to

unvaccinated controls (Fig. 5B,C). This difference is highly significant in 4-week-old pups ($p < 0.0001$) and still maintained when pups are 10 weeks of age ($p = 0.0014$). Quantification of total IgG verified that the difference between MatAbs in vaccinated and unvaccinated pups was rotavirus-specific (Fig. EV3A). No maternal IgA was transferred to the pup circulation (Fig. 1D), and quantification of total IgA showed no difference between groups (Fig. EV3B). We propose that this data supports the hypothesis that vaccine clearance is a key mechanism by which interference occurs; when MatAbs bind to vaccine particles in the intestine, this results in clearance of both MatAbs and vaccine.

Next, we questioned the fate of rotavirus-specific MatAbs after vaccination. Antibodies in complex with antigen in circulation are typically cleared by innate immune cells such as phagocytes or degraded in lysosomes if not bound by FcRn, but in the gastrointestinal tract, we asked if MatAbs would simply be shed in stool. To investigate this, we quantified MatAbs present in stool collected from the intestinal tract of 9-day-old pups, with or without vaccination at 7 days of age. As shown in Fig. 5D, there was a trend towards more IgG MatAbs detected in stool samples if the pups were vaccinated, but this was not statistically significant. As expected, there were no rotavirus-specific antibodies in the stool of 9-day-old pups vaccinated in the absence of MatAbs, as there was insufficient time to generate a rotavirus-specific immune response following vaccination just 2 days earlier. We also quantified IgA in the same stool samples, which would be solely of breast milk origin (Fig. 5E). Interestingly, a significant difference between IgA MatAbs detected in stool in vaccinated and unvaccinated pups was evident. To evaluate whether this vaccination-associated increase in MatAb excretion continued in the week after vaccination, IgA was quantified in stool samples pooled from litters of vaccinated pups ($+/-$ MatAbs, same pups as in Fig. 5A) and additional litters of unvaccinated pups. As shown in Fig. 5F, there was a trend towards higher excretion of rotavirus-specific IgA MatAbs in the stool of pups that were vaccinated compared to those that were not, especially in the first few days after vaccination (IgG levels were below the lower limit of detection). Infection and inflammation are known to upregulate pIgR in the mammary gland via interferon γ (Rincheval-Arnold et al, 2002), so we asked whether the increase in rotavirus-specific IgA in stool after pup vaccination could be attributed to live vaccine transmission to dams and a subsequent increase in total milk IgA. As shown in Fig. EV3C, we found that total IgA in pup stool samples was consistent across all timepoints and groups. We sought to verify this finding by quantifying MatAbs present in the milk of dams. We observed an upward trend in rotavirus-specific IgA and total IgA in milk in the first week after pup vaccination in dams with rotavirus antibodies, but this was not statistically significant (Fig. EV3D). Moreover, the gradual rise in milk IgA MatAb titers does not correlate with the rapid increase in rotavirus-specific IgA titers detected in stool after pup vaccination. We suggest that this provides evidence that vaccination enhances clearance of rotavirus-specific IgA MatAbs via the gastrointestinal tract, whereas the fate of IgG MatAbs remains unclear.

To evaluate the long-term effects of MatAb interference on the ability of mice to mount an immune response to rotavirus infection after MatAbs had waned, we challenged naïve and vaccinated mice with a high dose of rotavirus (10 FFU) when MatAbs had waned at 10–12 weeks of age. Stool samples were collected daily from each mouse for quantification of virus shedding by viral-antigen ELISA, and then terminal serum samples were

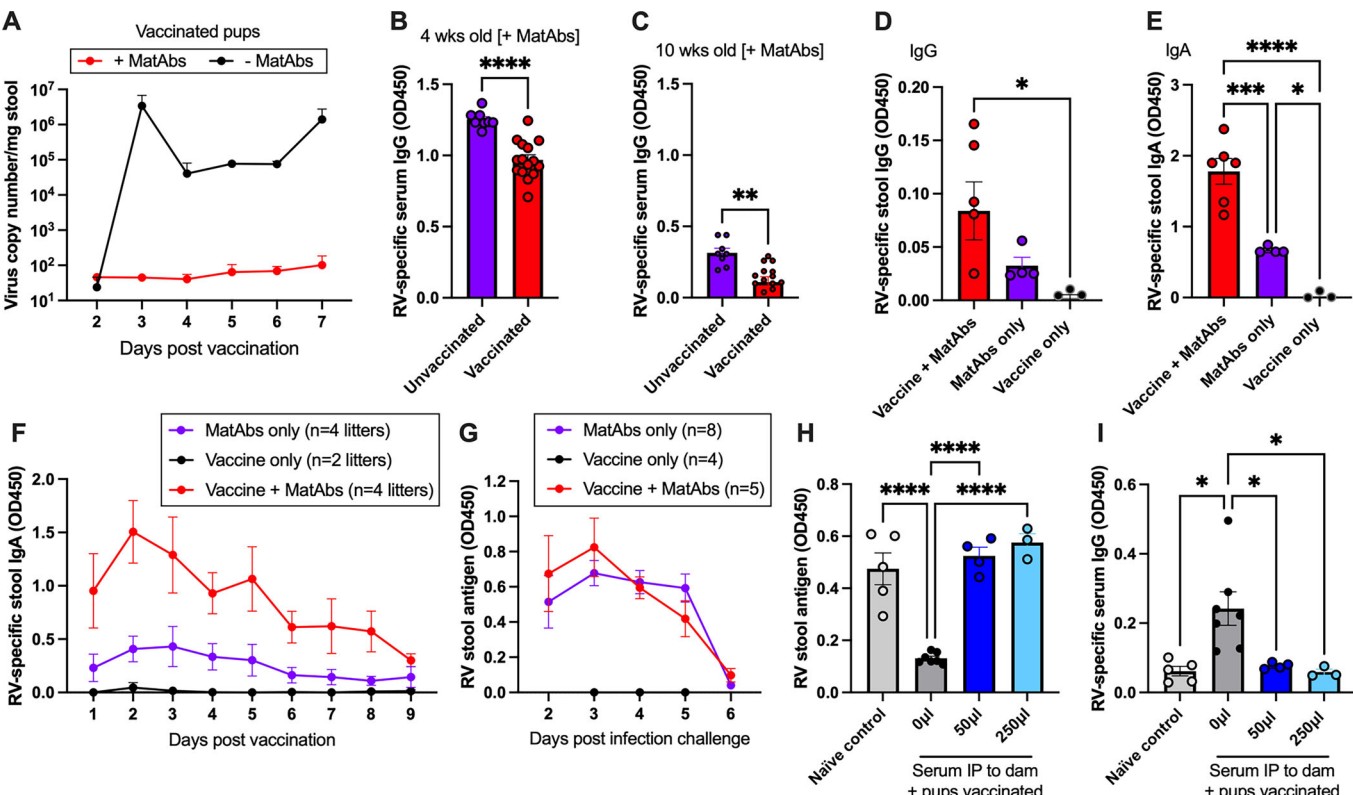

**Figure 5. Combined clearance of vaccine and maternal antibodies, with no long-term effects on immunity.**

(A) qPCR of rotavirus in stool samples collected after vaccination in pups with and without MatAbs ($n = 2$ litters each condition, samples from each litter pooled at each timepoint). (B, C) Rotavirus-specific IgG was quantified by ELISA in mice receiving MatAbs that did (2 litters, $n = 15$) or did not (1 litter, $n = 8$) receive a vaccine at 7 days old, in serum samples collected at 4 weeks ($p < 0.0001$) (B) or 10 weeks ($p = 0.001$) (C) of age. (D, E) IgG (D) or IgA (E) were quantified in stool samples by ELISA in 9-day old pups +/- rotavirus vaccination; $p = 0.02$ for vaccine + MatAbs vs vaccine only (D) and $p < 0.0001$, $p = 0.0002$, $p = 0.01$ for vaccine + MatAbs vs vaccine only, vaccine + MatAbs vs MatAbs only, and vaccine only vs MatAbs only, respectively (E); $n = 6$, $n = 4$, and $n = 3$ for vaccine + MatAbs, MatAbs only, and vaccine only, respectively. (F) Stool samples were pooled from each litter, and IgA was quantified by ELISA. (G, H) Adult mice born +/- MatAbs were challenged with rotavirus when 10–12 weeks of age; rotavirus shedding was quantified by viral-antigen ELISA in (G) mice born to naturally infected dams, and (H) mice born to dams receiving MatAbs IP; $p < 0.0001$, $p < 0.0001$, $p < 0.0001$ for 0 μL serum vs 50 μL, 250 μL, and naïve control, respectively; $n = 5$, $n = 7$, $n = 4$, and $n = 3$ for naïve control and 0, 50, and 250 μL serum, respectively. (I) Rotavirus-specific IgG induced or boosted by challenge as quantified 2 weeks post-infection; $p = 0.04$, $p = 0.03$, $p = 0.01$ for 0 μL serum vs 50 μL, 250 μL, and naïve control, respectively; $n = 5$, $n = 7$, $n = 4$, and $n = 3$ for naïve control and 0, 50, and 250 μL serum, respectively. Data information: For all graphs, data were presented as mean, and error bars represent the standard error of the mean. Statistical significance was determined by unpaired two-tailed $t$-test (B, C), one-way ANOVA (D–F, H, I), or two-way ANOVA with repeated measures (F, G). Significant Tukey's adjusted pair-wise comparisons (F, G) and Bonferroni-corrected pair-wise comparisons (D, E, H, I) are shown (*$p \leq 0.05$; **$p < 0.01$, ***$p < 0.001$, ****$p < 0.0001$). Source data are available online for this figure.

collected 2 weeks later. As shown in Fig. 5G, virus shedding in mice that had received MatAbs as pups from naturally infected dams was identical between pups vaccinated at 7 days of age and unvaccinated mice. To confirm our findings, we also challenged mice that were born to dams who received MatAbs by passive transfer (as in Fig. 3E,F). We found that virus shedding of stool antigen was the same in naïve control adult mice as compared to two litters of mice that were vaccinated in the presence of MatAbs, as shown 4 days post infection in Fig. 5H. Examination of antibody responses 2 weeks after virus challenge identified comparable serum IgG titers induced in naïve and MatAb groups (Fig. 5I), and no differences in stool IgA across all groups (Fig. EV2B).

## Single-cell analysis reveals interference with the global anti-viral response

Whilst our results had so far proven the ability of MatAbs to interfere with germinal center formation and class-switched

antibody production, the effect of MatAbs on individual immune cell responses were not well defined. To address this, we comprehensively profiled immune diversity at the single-cell level of the draining mesenteric lymph node (MLN). Cluster analysis of all cells, a total of 31,791 cells from the vaccine + MatAbs (15,334 across two biological replicates) and vaccine only (16,457 across two biological replicates) conditions, identified 18 distinct clusters (Fig. EV4B) and canonical markers annotated the majority of clusters (Fig. 6B–D). B cells were the most abundant cell type (39 vs 36% for vaccine + MatAb and vaccine only, respectively, $p = 0.5$) in all samples. Differences in immune cells between conditions were limited, although differences were identified for activated CD8 T cells (1.2 vs 3.2% for vaccine + MatAb and vaccine only, $p = 0.06$), cytotoxic CD8 T cells (1.8 vs 2.4% for vaccine + MatAb and vaccine only, $p = 0.05$) and pDC (0.17 vs. 0.36% for vaccine + MatAb and vaccine only, $p = 0.03$). Marginally decreased activated and cytotoxic CD8 T cells at 10 days post vaccination is consistent

with the absence of viral replication and shedding in MatAb mice following vaccination.

Given the robust failure of pups to seroconvert following vaccination in the presence of MatAbs, we were specifically interested in identifying transcriptome changes in the MatAb group and used a contrast to compare vaccine + MatAb and vaccine only in differential expression analysis. We hypothesized naïve pups would mount the requisite immune response to vaccination that would result in differentiation of antibody-secreting cells, and any changes observed in the MatAb group would be associated with the failure to seroconvert. We further hypothesized that if vaccine clearance is a predominant mechanism of interference, there would be limited evidence of immune activation in lymphocytes in the draining lymph node. Our approach identified individual genes with expression differences between vaccine + MatAb and vaccine only for cell types with greater than 282 cells per cluster, but not for less abundant cell types (Fig. 6E). Of note, more downregulated genes were identified in the MatAb group across altered cell types.

To characterize the coordinated shifts in gene expression reflecting changes in pathway activation between vaccine + MatAb and vaccine only, we performed gene set enrichment analysis (GSEA) on each cluster in which we detected differential gene expression (Fig. EV5A). The most substantial finding from this analysis was the downregulation of interferon responses, in particular, gene sets associated with the hallmark interferon alpha and interferon gamma response were downregulated in nearly all cell types of interest (Fig. 6F), including activated CD4 T cells, which were downregulated in all top 10 core enrichment genes (Fig. EV5B–D). We further investigated a subset of interferon-stimulated genes (ISGs) previously associated with rotavirus infection in enterocytes in mice (Bomidi et al, 2021). As expected, the expression of selected ISGs was lower in vaccine + MatAb compared to vaccine only (Fig. 6G). These results demonstrate a robust and global decrease in the transcriptome that drives response to vaccination.

To further investigate the effect of MatAbs on B cell populations in the MLN, we focused on B cell lineage in a separate analysis (Fig. 7A). The detectable expression of the joining chain of multimeric antibodies (Jchain), the marker for somatic hypermutation and class switch recombination (Aicda), and proliferation markers (Stmn1, Mki67) clearly identified a cluster of plasma and germinal center (GC) B cells undergoing active proliferation and differentiation into antibody-secreting cells (Fig. 7B). The proportion of B cell types was not altered except for a marginal increase in the naïve – 1 cluster ($p = 0.03$) in vaccine + MatAb compared to vaccine only, 68.1 vs. 63.8%, respectively, and a decrease ($p = 0.03$) in plasma and GC B cells in vaccine + MatAb compared to vaccine only, 0.64 vs. 1.65%, respectively (Fig. 7C). Along with the decrease in plasma and GC B cells, it followed that counts for differentiation markers for this population were lower or undetected but that Ighd was greater in vaccine + MatAb compared to vaccine only (Fig. 7D). Taken together, these data confirm that Ig class switching, cell proliferation, and initiation of plasma cell differentiation were limited in the presence of MatAbs. Differential expression and GSEA further revealed an altered B cell transcriptome (Fig. EV5E,F), including a downregulated interferon response (Fig. EV5G).

In conclusion, we observed a blunted interferon response and diminished B cell activation and differentiation in the presence of MatAbs. These results support the hypothesis of reduced antigen encounter and vaccine clearance by MatAbs that blunted the antibody response to vaccination.

## Methods to overcome maternal antibody interference

The predominant strategy used to reduce the negative effect of MatAbs on human infant responses to vaccines is to delay vaccination until MatAbs are expected to have sufficiently waned (Pollard and Bijker, 2021; Niewiesk, 2014). We asked whether delaying vaccination would permit pups to seroconvert to rotavirus vaccination in our experimental model. Instead of vaccinating at 7 days old (modeling 8 weeks old in human infants), we vaccinated at weaning (3 weeks old) or at 4 weeks old when breast milk MatAbs are no longer entering the gastrointestinal tract. As shown in Fig. 8A, delaying vaccination in our system still resulted in robust MatAb interference. This indicates that circulating antibodies in the pup alone are sufficient to interfere with immunity in mice.

Given a clear role for MatAb-mediated vaccine clearance in our model, we predicted that there must be a vaccine dose whereby MatAbs are out-competed and seroconversion can take place. Keeping MatAb titers the same, we immunized different litters of pups with increasing doses of vaccine, and monitored for seroconversion after weaning. Figure 8B shows that increasing the vaccine dose simply increased the rate at which MatAb waning occurred. We were unable to identify a threshold over which vaccine dose induced seroconversion.

Next, we sought to determine if a different vaccine type could overcome MatAb interference. To study this, we chose to compare non-replicating subunit vaccines with the standard live-attenuated vaccine via the same oral delivery route. We immunized litters of pups with a single rotavirus protein, the inner capsid protein VP6, in the presence or absence of MatAbs. VP6 is known to be highly conserved and highly immunogenic, and has been considered as a vaccine target by multiple groups (Afchangi et al, 2019). We studied two different formats of VP6; either purified recombinant protein (kind gift from James Crowe), or in the form of a non-infectious double-layered particle (DLP), purified from infected cells by ultracentrifugation as previously described (Caddy et al, 2020). Pups were orally vaccinated at 7 days old with 10 µg of each preparation, and blood samples were collected at 4 to 10 weeks of age as standard. As shown in Fig. 8C, when week 10 rotavirus-specific IgG was quantified by ELISA, recombinant VP6 was not immunogenic in any pups. In contrast, DLPs induced seroconversion in pups to a comparable level in pups with or without MatAbs. This indicated that MatAbs were not able to sufficiently clear DLPs and evade activation of neonatal B cells. However, DLPs alone were not as immunogenic in all pups as the live-attenuated vaccine approach.

Finally, we asked whether the route of vaccine delivery could be modified to improve vaccine efficacy in the presence of MatAbs. Vaccine administration via different routes has previously been shown to be useful in reducing MatAb interference (Yang et al, 2019; Johansson et al, 2008). We chose to compare parenteral delivery with oral delivery, predicting that parenteral administration could improve vaccine efficacy. For this experiment, we also included adjuvants as parenteral administration alone is known to have poor immunogenicity (Zhang et al, 2015). DLPs were mixed

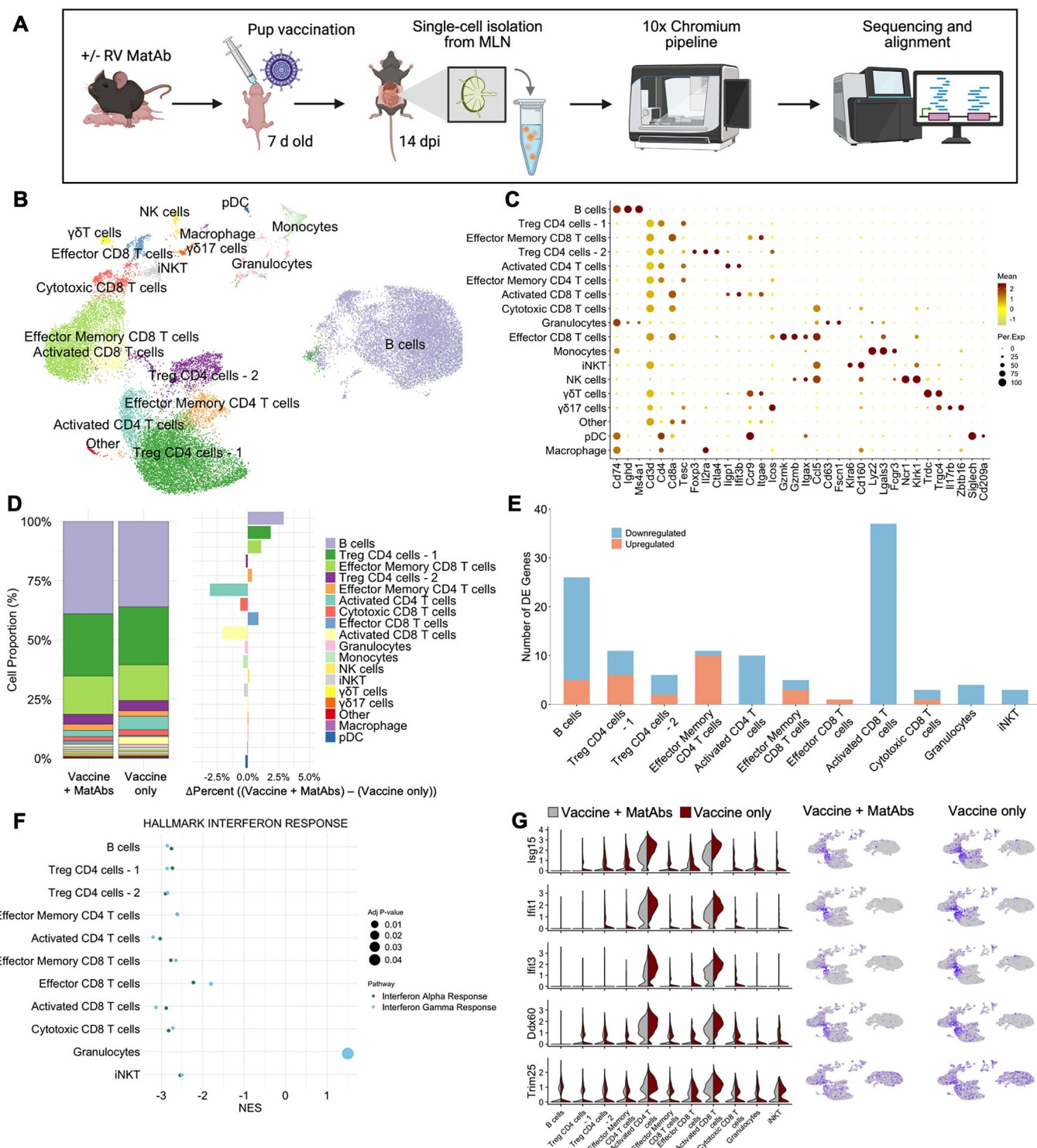

**Figure 6. Single-cell transcriptomics during maternal antibody interference.**

(A) Schematic diagram of experimental protocol. (B) Integrated uniform manifold approximation and projection (UMAP) for all samples. Individual cells plotted. (C) Relative expression of canonical and marker genes (x-axis) for immune cells across clusters (y–axis); dots indicate average expression and percentage of cells detected with expression (color and size, respectively). (D) Average proportion of cell types in vaccine + MatAb and vaccine only samples (left) and their change in proportion between vaccine + MatAb and vaccine only samples (right). (E) Counts of differentially expressed (DE) genes (y-axis) per cell type, comparing vaccine + MatAb and vaccine only with absolute log2 fold change >1 and adjusted p value ≤0.05. (F) GSEA results showing downregulation of hallmark interferon alpha and gamma response in major clusters. Dots are sized to denote significance; the x-axis indicates NES. (G) Distribution and expression of selected interferon-stimulated genes from mesenteric lymph nodes isolated from pups vaccinated in the presence (vaccine + MatAbs, n = 2) or absence of MatAbs (vaccine only, n = 2). Source data are available online for this figure.

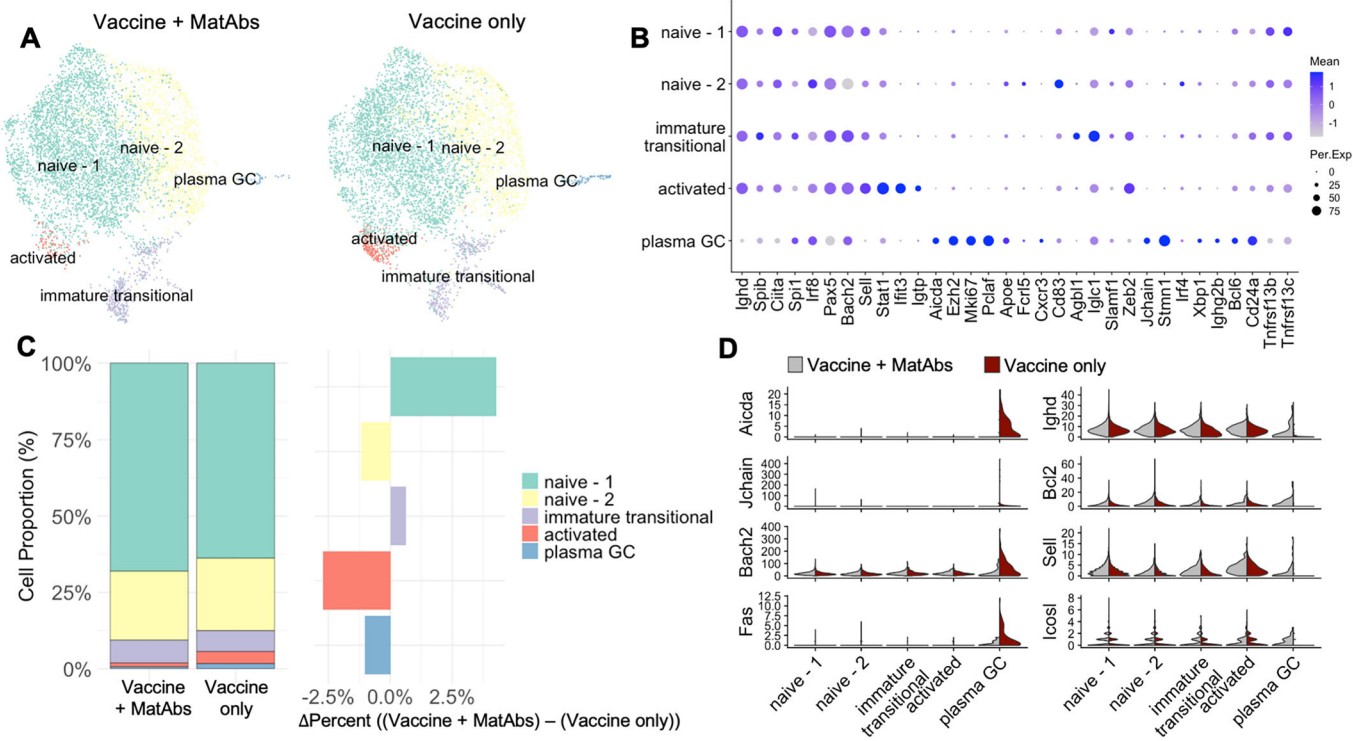

**Figure 7. B cell transcriptomics during maternal antibody interference.**

(A) Integrated uniform manifold approximation and projection (UMAP) for vaccine + MatAb and vaccine only samples. Individual cells plotted. (B) Relative expression of canonical and marker genes (x-axis) for immune cells across clusters (y-axis); dots indicate average expression and percentage of cells detected with expression (color and size, respectively). (C) Average proportion of cell types in vaccine + MatAb and vaccine only samples (left) and their change in proportion between vaccine + MatAb and vaccine only (right). (D) Distribution and counts of selected B cell differentiation markers from mesenteric lymph nodes isolated from pups vaccinated in the presence (vaccine + MatAbs, $n = 2$) or absence of MatAbs (vaccine only, $n = 2$). Source data are available online for this figure.

with Addavax (InvivoGen, MF59-like) for parenteral delivery, or cholera toxin B (CTB, Sigma-Aldrich) for oral delivery, as Addavax is not recommended for oral administration. Figure 8D presents longitudinal rotavirus-specific IgG responses in mice vaccinated in the presence of MatAbs. Whilst oral vaccine delivery with CTB was less immunogenic than expected, the majority of pups overcame MatAb interference by parenteral administration of DLPs to mount a robust IgG response. Intriguingly, no mice made an IgA response to either vaccine route in the presence of MatAbs (Fig. 8E), again demonstrating how class switching can be influenced by MatAbs. To investigate the protective efficacy of this IgG-focused response, we quantified the ability of serum IgG from a subset of mice to neutralize virus in vitro. DLP-specific IgG cannot neutralize virus extracellularly as it does not target the outer capsid proteins, but DLP-specific IgG can be quantified via intracellular neutralization assays as previously described (Caddy et al, 2020; Woodyear et al, 2024). We showed that IgG induced by parental administration was able to mediate significantly more intracellular neutralization of rotavirus than orally administered vaccine (Fig. 8F). Finally, we challenged the same mice with a heterologous murine rotavirus when they were 12 weeks of age. As shown in Fig. 8G, viral shedding as quantified by ELISA on stool samples collected on day 4 post infection was significantly lower in mice with MatAbs vaccinated parenterally compared to those vaccinated orally.

## Discussion

Inducing protective immune responses in neonates is a major issue due to the dual challenge of immaturity of the neonatal immune system and MatAb inference (Demirjian and Levy, 2009; Hodgins and Shewen, 2012). However, despite the phenomenon of MatAb interference first being reported in 1958 (Perkins et al, 1958), the fundamental mechanisms responsible for this problem have been difficult to unravel. The need for suitable experimental models to study the effects of MatAbs on neonatal immune responses has long been sought after (Hodgins and Shewen, 2012; Niewiesk, 2014). Here, we have presented our solution to understanding how an orally delivered live-attenuated vaccine can be blocked by MatAbs. We have focused on rotavirus as a significant global health concern, and demonstrated that MatAbs are highly efficient at clearing rotavirus vaccines from the gastrointestinal tract in our mouse model. We propose that this contributes to the poor seroconversion observed in low-and middle-income countries to current rotavirus vaccines.

A small number of earlier studies have examined the effect of MatAbs on different types of rotavirus vaccines in mice (Yang et al, 2019; Johansson et al, 2008; Zhou et al, 2022). Yang et al, 2019, used a low dose of the live murine rotavirus of the EDIM strain as the vaccine and demonstrated MatAb interference with seroconversion in offspring. Johansson et al, 2008 and Zhou et al, 2022 used non-live vaccine approaches (triple-layered virus-like particles and

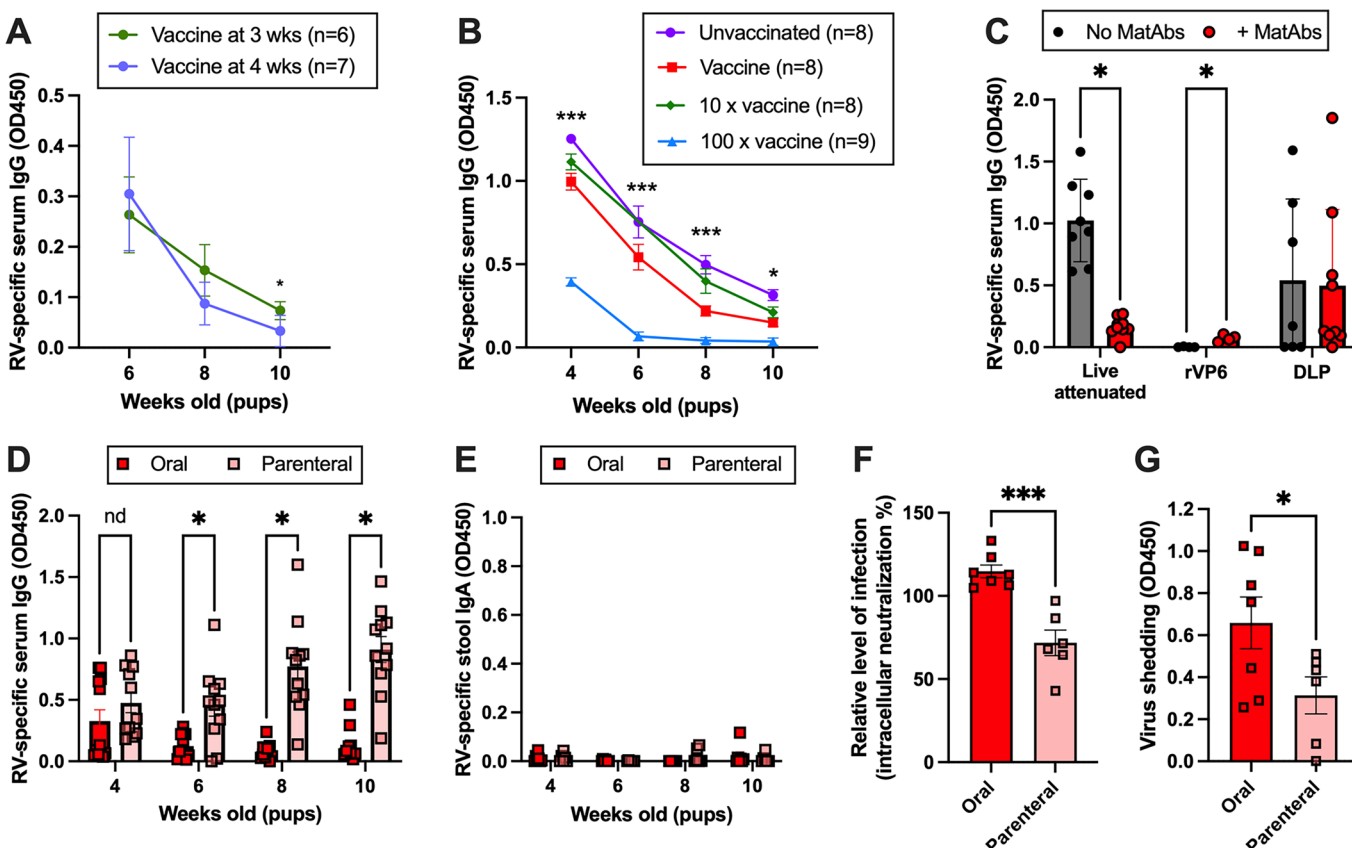

**Figure 8. Overcoming maternal antibody interference.**

(A) Rotavirus-specific IgG in the serum of pups with MatAbs that were vaccinated at 3 or 4 weeks old; $p = 0.05$ at 10 wk. (B) Serum IgG ELISA of litters of pups that received MatAbs, which were unvaccinated or vaccinated with increasing doses of the rotavirus vaccine at 7 days old; $p < 0.0001$, $p < 0.0001$, $p = 0.0004$, and $p = 0.01$ at 4, 6, 8, and 10 wk, respectively, for vaccine vs 100x vaccine. (C) Pups born to seronegative (no MatAbs) or seropositive (+ MatAbs) dams were vaccinated at 7 days old with the standard live-attenuated vaccine ($p < 0.0001$), recombinant VP6 (rVP6) ($p = 0.007$), or purified DLPs. Presented is serum rotavirus-specific IgG when pups were 10 weeks old; $n = 8$, $n = 4$, and $n = 7$ for live-attenuated, rVP6, and DLP in No MatAbs, respectively, and $n = 9$, $n = 4$, and $n = 9$ for live-attenuated, rVP6, and DLP in + MatAbs, respectively. (D) Longitudinal serum IgG ELISA of pups vaccinated in the presence of MatAbs with DLPs delivered orally with cholera toxin adjuvant (2 litters, $n = 12$) or administered parenterally with Addavax adjuvant (2 litters, $n = 11$) at 4, 6 ($p = 0.03$), 8 ($p = 0.03$), and 10 ($p = 0.02$) wk post vaccination. (E) Longitudinal stool IgA ELISA of pups from (D). (F) Intracellular neutralization of rotavirus by serum antibodies from a subset of pups that received oral ($n = 7$) or parenteral ($n = 6$) vaccination in (D) when 10 weeks old ($p = 0.0003$). (G) Viral shedding as quantified by viral-antigen ELISA from stool samples collected 4 days after challenge when pups were 12 weeks old ($p = 0.05$) following oral ($n = 7$) or parenteral ($n = 6$) vaccination. Data information: Statistical significance was determined by two-way ANOVA with repeated measures (A–C) and unpaired two-tailed *t*-tests (C, F, G). Significant Bonferroni-corrected pair-wise comparisons (A, C, D) and Tukey's adjusted pair-wise comparisons between vaccine and 100x vaccine (B) are shown (*$p \leq 0.05$; **$p < 0.01$, and ***$p < 0.001$; error bars indicate SEM. Source data are available online for this figure.

inactivated human rotavirus strain, respectively) and also both observed MatAb interference. The robust MatAb interference observed in these studies and our model is unexpectedly distinct from the results presented in a recent pre-print by Muleta et al (Muleta et al, 2024). Interestingly, in that study, it was reported that humoral immunity was still induced in the presence of protective MatAbs, albeit delayed until after weaning. This is in contrast to our findings, where no humoral immunity was observed for at least 7 weeks after weaning. A key difference between these studies is that Muleta et al challenged pups with a very high dose of the virulent murine rotavirus strain ECw, whereas in our study, we modeled vaccination using low doses of the attenuated murine rotavirus strain EMcN. We therefore propose that our model is more reflective of MatAb interference with rotavirus vaccination, whereas Muleta et al demonstrate interesting findings that align with potent natural infection.

A role for FcγRIIB in driving IgG-mediated interference has long been considered in a variety of different immunological contexts. Our results in an FcγRIIB KO mouse line have shown that FcγRIIB is not central to MatAb interference to rotavirus vaccination when MatAb titers are induced by natural infection. This is in agreement with a previous study that examined immune responses to sheep erythrocytes (Karlsson et al, 1999). When anti-sheep erythrocyte IgG was passively transferred to mice at the same time as sheep erythrocytes, no anti-sheep erythrocyte IgG was induced in either wildtype or FcγRIIB KO mice. However, we have shown in passive transfer experiments with very low MatAbs titers that whilst pup IgG responses are still prevented in wild type mice, the absence of FcγRIIB can permit the induction of pup IgG responses. These results align with previous studies using a cotton rat model to study MatAb-mediated interference with measles vaccination (Kim et al, 2011). In this study, F(ab')2 fragments

passively transferred at the same time as vaccination did not suppress neutralizing antibody production. It was concluded that this indicated that the Fc region of IgG was required for suppression. Furthermore, only IgG isotypes capable of binding to FcγRIIB were able to mediate suppression. Being a rat model of vaccination, no FcγRIIB KO animals were available, and results are therefore largely indirect, so it is valuable that we have verified these findings in KO mice. Our discovery that systemic pup IgA responses continued to be inhibited in FcγRIIB KO mice is of particular interest, as previous studies have not evaluated IgA titers in this context. Our results provide evidence that MatAbs can interfere with pup IgG and IgA responses via different mechanisms, demonstrating a level of redundancy. Whilst it might be expected that IgG and IgA MatAbs can impact the neonatal immune responses in different ways, future work needs to identify the mechanisms by which neonatal B cell class switching can be influenced by MatAbs. In addition, the effects of MatAbs on the earliest stages of the humoral response, namely IgM production, remain poorly understood. Investigating IgM responses at early timepoints post vaccination could provide further insight into the initiation of neonatal immunity in the presence of MatAbs.

Currently licensed rotavirus vaccines in human infants are all live-attenuated vaccines, and replication in the neonatal gastro-intestinal tract is understood to be important for inducing an optimum vaccine response (Burke et al, 2024; Lee et al, 2021). Our results provide strong evidence for MatAb-mediated clearance of vaccine particles, which prevents the neonatal B cell from encountering their target. This is in agreement with another recently reported study focusing specifically on breast milk antibodies to SARS-CoV-2 in a mouse model (Dangi et al, 2024). Furthermore, we observed significantly reduced IgG MatAbs in pup circulation after vaccination. We suggest this process can be described as 'antibody consumption' (Tas et al, 2022), but the fate of IgG MatAbs after vaccination remains uncertain. We investigated whether IgG MatAbs would be cleared from the gastro-intestinal tract in stool, but only observed a slight increase in IgG detected in stool post vaccination. It is possible that IgG bound to the vaccine is cleared by antibody-mediated phagocytosis (Siegrist, 2003); we were unable to detect any differences in phagocyte populations in our single cell RNA-sequencing analysis, but this could be due to the low number of phagocytes captured, or the location in which phagocytes mediate such activity. Importantly, however, we did observe significantly increased excretion of IgA MatAb in stool samples following vaccination. We demonstrated that this could not be solely attributed to increased delivery of IgA due to interferon-γ-induced upregulation of pIgR in the mammary gland, as rotavirus-specific IgA in pup stool increased much more rapidly than total IgA in milk. This suggests that IgA MatAbs in milk are binding to the incoming vaccine and then being cleared by the gastrointestinal tract. It also highlights that there are likely some differences in how IgG and IgA MatAbs can mediate interference in our mouse model, which could be further dissected using systems to study each isotype independently. Finally, we reasoned that if vaccine clearance was the primary mechanism of interference, then the response to rotavirus infection when MatAbs had waned would be the same in naïve versus vaccinated + MatAb mice. We were able to confirm this in challenge experiments that demonstrated MatAbs had no lasting impact on the ability of mice to respond to rotavirus infection.

Our single-cell RNA-sequencing analysis has provided a detailed insight into the cell population and gene expression changes that occur in MLN after live-attenuated rotavirus vaccination in the presence and absence of MatAbs. Our overarching finding was that vaccination of naïve neonatal mice induces a robust set of changes in anti-viral immune pathways, which are still readily detected two weeks post vaccination. In the presence of MatAbs, there was a significant decrease in activation of any of these pathways, which is in agreement with our more focused experimental approaches. Our study is the first to report single-cell analysis of the lymph node response to rotavirus infection, and as such, provides valuable insight into the anti-viral pathways stimulated. Previous work has shown that significant interferon-related pathways are induced in intestinal epithelial cells where CD45+ cells were removed from sample processing (Bomidi et al, 2021), so our study provides useful complementary results.

A limitation of using a mouse model to study MatAb transfer is the biological differences in neonatal immune development and MatAb transfer mechanisms relative to humans. Mice are born more developmentally immature than human infants (Veru et al, 2014), but still capable of mounting an IgG response from birth (Kolb et al, 1974). Whilst both humans and mice have a hemochorial placenta, making mice a good model in comparison to species with different placentation types, significantly fewer MatAbs are delivered transplacentally in mice compared to humans. Whereas the human placenta permits transfer of the majority of IgG, in mice, only 30% of maternal IgG is found in pups at birth (Appleby and Catty, 1983). To make up for this reduced transfer pre-birth, mouse milk contains significantly higher titers of IgG than human milk. However, we have largely focused our study on seroconversion from 4 weeks of age (1-week post weaning), so MatAb transfer will be complete irrespective of the route of delivery. Throughout most of our study, we therefore considered the impact of total MatAb transfer as a whole, and did not divide our conclusions into breast milk versus transplacental antibodies, nor IgG versus IgA. We had two interesting exceptions to this approach though; delivery of artificial MatAbs in the form of pooled serum administered IP to dams, and vaccination after breastfeeding had ceased. These experiments allowed us to show that in mice, breast milk antibodies alone can mediate interference, and also that MatAbs in the pup circulation (which will have originated from both transplacental and milk delivery) can be sufficient to mediate interference after weaning. An additional limitation to our study is that we have specifically focused on how MatAbs influence neonatal immunity, without broader consideration of immune cells and cytokines that can be transferred from mother to offspring. These additional factors can have a significant role in shaping neonatal immune responses (Fernandes and Lim, 2024; Jennewein et al, 2017), and it will be valuable to build an evaluation of these into future studies.

A key motivator for understanding the biological mechanisms underpinning MatAb interference is the potential for rational development of vaccine strategies to overcome this issue. Although delaying vaccination for longer to permit more MatAb waning has shown potential in human infants (Church et al, 2019), this strategy had no benefit in our mouse model, and in heterogenous human populations, this certainly risks leaving some infants

vulnerable for longer if MatAb delivery was impaired in any way. Increasing vaccine dose has been considered as an option for evading MatAb interference, and has shown promise in some human rotavirus clinical trials (Appaiahgari et al, 2014). However, this approach was ineffective in our model, for reasons that remain unclear but could be due to the vaccine dose to MatAb ratio. It is important to consider that increasing the vaccine dose is also expected to increase the risk of side effects, which have challenged the rotavirus vaccine field since the withdrawal of the first licensed rotavirus vaccine in 1999 (Centers for Disease Control and Prevention (CDC), 1999). Development of subunit rotavirus vaccines for parenteral administration has been a significant avenue of research in recent years (Song, 2021). Our results are in agreement with two other murine studies that demonstrated parenteral administration of rotavirus vaccines could overcome interference more effectively than oral vaccine administration (Yang et al, 2019; Johansson et al, 2008). We demonstrated that vaccination via the parenteral route resulted in seroconversion in the majority of mice within a cohort, and that this correlated with reduced virus shedding when these mice were challenged with rotavirus as adults. These results therefore support development of a parenteral rotavirus vaccine as a strategy to minimize the negative effects of MatAbs, although the difficulties in inducing mucosal immune responses via this route still require significant attention (Correa et al, 2022). It is clear from the results of the recent phase III vaccine trial with a parenteral non-replicating P2-VP8 vaccine that additional challenges remain (PATH, 2024). This parenterally delivered vaccine was proven to be less effective at preventing severe gastroenteritis than the standard oral live-attenuated vaccine. Following this, there is considerable interest in developing alternative vaccine platforms that will be effective irrespective of MatAbs. Several mRNA-based rotavirus vaccines have been reported in pre-clinical studies (Roier et al, 2023; Hensley et al, 2024; Lu et al, 2024), which could be promising, although mRNA-based vaccines are partially affected by MatAbs for both influenza and SARS-CoV in pre-clinical vaccine studies (Schumer et al, 2024; Willis et al, 2020).

In summary, we have shown that MatAb-mediated clearance of vaccine particles is a predominant mechanism by which MatAbs interfere with the ability of the neonate to respond to oral live-attenuated rotavirus vaccines in mice. We propose that understanding this biological phenomenon will be valuable for informing future vaccine development.

## Methods

### Reagents and tools table

| Reagent/resource | Reference or source | Identifier |
| --- | --- | --- |
| **Experimental models** | | |
| MA104 cells (*C. sabaeus*) | ATCC | |
| C57BL/6 (*M. musculus*) | JAXTM | N/A |
| BALB/c (*M. musculus*) | JAXTM | N/A |
| FcγRIIB KO (*M. musculus*) | Gift of J. Ravetch, Rockefeller, USA | N/A |

| Reagent/resource | Reference or source | Identifier |
| --- | --- | --- |
| **Antibodies and dyes** | | |
| Anti-Ki-67-FITC | BD Pharmingen | CAT#558616; Clone: B56 |
| Anti-mouse IgA HRP (Goat) | Bio-Rad | CAT#STAR137P |
| Anti-mouse IgG HRP (Goat) | Sigma-Aldrich | CAT#A0168 |
| Anti-IgG1a Biotin (Mouse) | BD Pharmingen | CAT#553500; Clone: 10.9 |
| Anti-rabbit IgG HRP (Goat) | BD Pharmingen | CAT#554021 |
| APC anti-mouse CD95 (Fas) | BioLegend | CAT#152603; Clone: SA367H8 |
| Armenian Hamster anti-PD-1 | Novus Biologicals | CAT#NBP1-43110AF700 |
| CD45 (PE Rat Anti-Mouse CD45R/B220) | BD Pharmingen | CAT#553090; Clone: RA3-6B2 |
| CD45R (PerCP/cyanine 5.5 anti-mouse/human CD45R/B220) | BioLegend | CAT#103236; Clone: RA3-6B2 |
| Fc Block (TruStain FcX anti-mouse CD16/32) | BioLegend | CAT#101320; Clone: 93 |
| FITC anti-mouse CD185 (CXCR5) | BioLegend | CAT#145520; Clone: L138D7 |
| GL7 Antigen (Pacific Blue Anti-Mouse/Human) | BioLegend | CAT#144614; Clone: GL7 |
| Rotavirus Polyclonal Antibody (Rabbit) | Antibodies-online | CAT#ABIN308233 |
| Rotavirus Polyclonal Antibody (Sheep) | Invitrogen | CAT#PA1-85845 |
| **Chemicals, enzymes, and other reagents** | | |
| Fixable Viability Dye eFlour 780 (L/D) | Invitrogen | CAT#65-0865 |
| Hoechst 33342 | Invitrogen | CAT#H3570 |
| Streptavidin-HRP | Mabtech | CAT#3310-9 |
| Dulbecco's Modified Eagle's Medium (DMEM) | Corning | CAT#10-013 |
| Endotoxin-Free Dulbecco's PBS (1X) | EMD Millipore | CAT#TMS-012-A |
| Fetal Bovine Serum (FBS) | Corning | CAT#28622001 |
| Oxytocin | Bimeda | CAT#1OXY015 |
| Pierce Protease Inhibitor Mini Tablets | Thermo Fisher Scientific | CAT#A32953 |
| RIPA Lysis and Extraction Buffer | Thermo Fisher Scientific | CAT#89900 |
| Sudan Black B | Ward's Science | CAT#470109-394 |
| TPCK-Treated Trypsin | Worthington Biochemical | CAT#LS003740 |
| Triton X-100 | MB Biomedicals | CAT#807423 |
| Xylenes | Thermo Fisher Scientific | CAT#L13317.AP |
| 3,3′,5,5′-Tetramethylbenzidine (TMB) Membrane Peroxidase Substrate Plus | Avantor | CAT#K830 |
| **Other** | | |
| BCA Protein Assay Kit | Thermo Fisher Scientific | CAT#23227 |

| Reagent/resource | Reference or source | Identifier |
|---|---|---|
| Luna Universal One-Step RT-qPCR Kit | New England Biolabs | CAT#E3006 |
| Monarch Total RNA Miniprep Kit | New England Biolabs | CAT#T2010S |
| BioTek Cytation 7 Cell Imaging Multimode Reader | Agilent | |
| Neon ® Transfection System | (Thermo Fisher Scientific) | |
| Illumina NovaSeqX | Illumina | |
| Deposited single-cell RNA-seq data | | [GEO Accession]: [GSE291166] |
| **Software** | | |
| Gen5 Image Prime (v3.13) | https://www.biotek.com/products/software-robotics-software/gen5-microplate-reader-and-imager-software/ | RRID:SCR_017317 |
| GraphPad Prism (v10.3.1) | https://www.graphpad.com | RRID:SCR_002798 |
| R (v4.4.1) | https://www.r-project.org | RRID:SCR_001905 |
| Cellranger (v8.0.0) | 10x Genomics | RRID:SCR_017344 |
| Seurat (v5.2.1) | https://doi.org/10.1038/s41587-023-01767-y | RRID:SCR_016341 |
| DEseq2 (v1.44.0) | https://doi.org/10.1186/s13059-014-0550-8 | RRID:SCR_015687 |
| clusterProfiler (v4.12.16) | https://doi.org/10.1089/omi.2011.0118 | RRID:SCR_016884 |
| fgsea (v1.20.0) | https://doi.org/10.1101/060012 | RRID:SCR_020938 |
| MsigDB database (v7.5.1) | https://doi.org/10.1073/pnas.0506580102 | RRID:SCR_016863 |
| FlowJo | BD Bioscience | RRID:SCR_008520 |

## Methods and protocols

### Mice

C57BL/6 (The Jackson Laboratory), BALB/c (The Jackson Laboratory), and FcγRIIB knockout (gift from J Ravetch, Rockefeller, USA) mice were housed and bred at the Baker Institute for Animal Health, Cornell University, or the Anne McLaren Building, University of Cambridge, under specific pathogen–free conditions.

### Virus preparation, infection, and vaccination

Murine rotavirus (EMcN) (McNeal et al, 2004) was generously provided by M McNeal, CCHMC, USA. The rotavirus vaccine strain (Rotarix) was grown in fetal monkey kidney (MA104) cells in the presence of trypsin, as previously described (Woodyear et al, 2024). Virus was trypsin-activated (10 µg/ml) at 37 °C for 30 min prior to MA104 cell infection. Virus preparations were titrated by fluorescent focus assay on MA104 cells and expressed as a number of fluorescent focus units (FFU), although replication of EMcN in vitro was limited. The virus was frozen at -80 °C until use. Prior to mouse infection or vaccination, the virus was diluted to the appropriate titer in sterile PBS without calcium chloride and magnesium chloride. Female mice were infected orally with 10 FFU

EMcN rotavirus (100 µl of 100 FFU/mL) or uninfected. Following confirmation of anti-rotavirus antibodies in the infected group, mating trios were established, and litters from seroconverted dams were randomized to receive vaccination or not. At 7 days of age, pups born to dams without (naïve) or with (MatAbs) anti-rotavirus antibodies were vaccinated with a dose of 0.01 FFU EMcN (50 µl of 0.2 FFU/mL) by oral gavage. Pups were monitored daily for diarrhea for 1 week following vaccination, and the endpoint was determined if pups were rejected by the dam. Purified recombinant VP6 (rVP6), provided by Dr. James Crowe (Vanderbilt Vaccine Center), and double-layered particles (DLP) purified from infected cells by ultracentrifugation as previously described (Caddy et al, 2020), were tested as an alternative vaccine type. Addavax (InvivoGen, MF59-like) and Cholera toxin B (CTB, Sigma-Aldrich) adjuvants were administered to compare an oral or parenteral vaccine delivery.

### Milk, serum, stool, and collection

To aid in the collection of milk, dams were anesthetized and received 2 IU/kg of oxytocin intraperitoneally prior to milking. Milk was aspirated from the mammary glands by manually expressing each teat. To collect serum, blood was collected through the lateral saphenous vein or through cardiac puncture, then centrifuged at $6000 \times g$ for 5 min, and the serum was stored at $-80$ °C. Stool samples from neonates were pooled daily from each litter and diluted 1:10 in PBS. Stool samples from adult mice were collected and stored without pooling. Diluted stool was centrifuged at $8000 \times g$ for 5 min to remove debris, and the resulting supernatant was stored at $-80$ °C.

### Passive transfer of the antibody mouse model

Serum from adult mice that had previously been infected with EMcN rotavirus was pooled, and rotavirus-specific IgG and IgA were quantified by ELISA. When pups were six days old, pooled serum diluted in endotoxin-free Dulbecco's PBS to a total volume of 250 µl was administered to dams IP. Twenty-four hours later, each pup in the litter was infected with EMcN by oral gavage. One week after the first passive transfer of pooled serum, a second dose of pooled serum at half the original volume was administered.

### IgG, IgA, and IgG1a ELISAs

Anti-rotavirus IgA, IgG, and IgG1a were determined by ELISA as previously described (Woodyear et al, 2024; Caddy et al, 2020) by an individual blinded to treatments. Briefly, rotavirus-specific polyclonal antibody (sheep) was plated at a concentration of 5 µg/ml in PBS. Purified cell culture lysates (virus-infected lysate or mock-infected control lysate) were diluted to 10 µg/ml in PBS. All samples were tested in duplicate at a 1:200 dilution (serum), 1:50 dilution (milk), 1:1000 dilution (pup stool), or a 1:100 dilution (adult stool) in 5% milk–PBS-T. IgG antibodies were measured using a goat anti-mouse IgG secondary antibody–HRP. IgA antibodies were measured using a goat anti-mouse IgA secondary antibody–HRP. IgG1a antibodies were measured using a mouse anti-IgG1a conjugated to biotin, followed by an additional 1-hour incubation with streptavidin-HRP diluted 1:1000 in 5% milk–PBS-T. TMB detected bound antibody, and the absorbance at 450 nm (OD450) was measured using a BioTek Cytation 7 Cell Imaging Multimode Reader, Gen5 Image Prime software (version 3.13).

### Antigen ELISA

Rotavirus-specific polyclonal antibody (sheep) was plated at a concentration of 5 μg/ml in PBS. Pup stool samples were tested at a 1:5000 dilution, and adult stool was tested at a 1:500 dilution, then incubated at 37 °C for 2 h. The anti-rotavirus polyclonal antibody (rabbit) was diluted 1:3000 in 5% milk–PBS-T and incubated at 37 °C for 1 h, followed by the anti-rabbit IgG-HRP (goat) under the same conditions. TMB was used to detect bound antibody, then read at 450 nm (OD450) on the BioTek Cytation 7 Cell Imaging Multimode Reader, Gen5 Image Prime software (version 3.13).

### Extracellular and intracellular neutralization assays

Antibody-mediated extracellular and intracellular neutralization assays were carried out as previously described (Woodyear et al, 2024). Briefly, for extracellular neutralization, MA104 cells ($2 \times 10^4$) were seeded in complete DMEM (Gibco) in a 96-well black sided plate (Corning 3340) and incubated at 37 °C for 4 h for cell adherence. Eight-twofold serial dilutions of serum in serum-free media (SFM) were incubated with trypsin-activated rotavirus at 37 °C for 1 h. The serum-virus mixture was then added in triplicate to seeded cells. After 1 h at 37 °C, 50 μl complete DMEM was added to each well and the plate was incubated at 37 °C for 16 h. Intracellular neutralization was carried out using the Neon® Transfection System (Thermo Fisher Scientific). Sera was first diluted 1:3 in PBS and mixed with $2 \times 10^5$ MA104 cells suspended in resuspension buffer R and electroporated with two pulses at 1400 V and a 20-pulse width. Electroporated cells were plated onto a 96-well black sided plate in complete DMEM. After incubation at 37 °C for 24 h, wells were washed once with PBS, and trypsin-activated rotavirus in SFM was added to each well. Infection proceeded as for the extracellular neutralization assay, described above. Rotavirus neutralization was quantified by fluorescent focus assay (FFA) for both assays. Quantification of rotavirus-infected cells was achieved using a BioTek Cytation 7.

### RT-qPCR

RNA was extracted from clarified stool suspensions using the Monarch Total RNA Miniprep Kit, following the manufacturer's instructions. Extracted RNA was denatured at 95 °C for 5 min to separate double-stranded RNA. RT-qPCR was performed using the Luna Universal One-Step RT-qPCR Kit, according to the manufacturer's instructions and containing the NSP5 sequences: forward primer CTGCTTCAAACGATCCACTCAC 400 nM, reverse primer TGAATCCATAGACACGCC 400 nM, TaqMan probe FAM-TCAAATGCAGTTAAGACAAATGCAGACGCT-TAMRA 200 nM. Amplification was performed on a QuantStudio 3 thermocycler (Applied Biosystems) with the following cycling conditions: 55 °C for 10 min, 95 °C for 1 min, followed by 40 × (95 °C for 10 s, 60 °C for 30 s). To quantify rotavirus genome copies per ml of stool supernatant, each plate included a tenfold serial dilution of SA11 total RNA. Samples with undetectable virus were assigned a lower limit of quantification of 100 genome copy numbers. Analysis was performed using QuantStudio Design & Analysis Software (version 1.5.1).

### Mesenteric lymph node single cell suspension

Mesenteric lymph nodes (MLNs) were harvested from infected mice. Single cells were isolated by mechanical disruption through a 70-μm cell strainer. Cells were washed once with RPMI + FBS, and resuspended in PBS + 1%FBS (flow cytometry) or PBS + 0.04%BSA (sequencing) before filtering through a 70–μm (flow cytometry) or 35-μm cell strainer (sequencing).

### Flow cytometry (FACS)

MLN single cells were incubated with Fc Block (1:100) in staining buffer (PBS + 1%FBS) for 30 min at 4 °C, then cells were incubated with viability dye (L/D) and fluorescently-conjugated antibodies targeting GC cells (GL7 Antigen and Fas) and TfH cells (PD-1 and CXCR5) in staining buffer for 30 min at 4 °C. Cells were resuspended in 100 μl 4%PFA-PBS at 4 °C for 15 min to fix, then resuspended in 300 μl staining buffer. Data were acquired using a BD LSRFortessa X-20 (BD Biosciences) and BD FACSDiva Software (version 9.0); analysis was performed using FlowJo (Treestar Inc., version 10.9.0).

### Single-cell counting and sequencing

MLN single cells were washed once with PBS + 0.04%BSA before cell counting. Single cell suspensions were run on a Chromium X instrument and libraries were prepared following the Chromium Next GEM Single Cell 3' RNA-Seq - Dual Index Assay version 3.1 (10x Genomics, user guide CG000315, RevE) by the Cornell BRC Genomics Facility (RRID:SCR_021727). We targeted 10,000 cells and used 13 cycles of cDNA amplification. Sample quality was confirmed using a Qubit (DNA HS kit; Thermo Fisher) to determine concentration and a Fragment Analyzer (Agilent) to confirm fragment size integrity. Libraries were sequenced on an Illumina NovaSeqX, 10B flowcell with $2 \times 150$ bp read length.

### Data processing

Fastq files were processed by the cloud-based cellranger count (version 8.0.0) available through 10x Genomics using default parameters and a murine reference (Mouse (GRCm39) 2024-A). Matrix files (min.cells = 3, min.features = 200) from each sample were imported into R for analysis with Seurat (version 5.2.1) (Hao et al, 2024). Initial filtering removed cells that did not meet minimum quality criteria [subset(sobj, subset = nFeature_RNA >1000 & nFeature_RNA <6000 & percent.mt <7.5 & log10GenesPerUMI >0.80)]. Additional parameters were calculated for further filtering using the PercentageFeatureSet, including percent.RBC ("^Hb[ab]-"), percent.platelet ("^Gypa"), and percent.CD45 ("^Ptprc"). Additional filtering was applied (percent.RBC <0.05 & percent.platelet <0.004 & percent.CD45 >0.003) to remove RBC, platelets, and cells with low Ptprc expression. To limit the effect of cell cycle score and ribosomal gene expression during downstream analysis, cell cycle scores (S.score, G2M.Score) and ribosomal gene scores (R.Score) were added to the matrix using CellCycleScoring and AddModuleScore using ribosomal genes, respectively. CellCycleScoring and AddModuleScore were calculated from the "scale.data" slot after LogNormalization and ScaleData were applied to the RNA assay. Cells with R.Score less than −2 were filtered out of the object.

Normalization of UMI counts for each sample was performed using the SCTransform (version 0.4.1) (Hafemeister and Satija, 2019; Choudhary and Satija, 2022) function [SCTransform(sobj, vars.to.regress = c("percent.mt", "S.Score", "G2M.Score", "R.Score"))]. Cell doublets within each sample were determined with scDblFinder (v1.18.0) (Germain et al, 2022) and removed from the dataset. Samples were integrated using the SeuratWrappers function

IntegrateLayers(method = HarmonyIntegration) (Korsunsky et al, 2019) (with default parameters. Clustering with Seurat [FindNeighbors(sobj, reduction = "harmony", dims = 1:50) %>% FindClusters(sobj, resolution = 0.6)] generated a total of 22 clusters. The FindMarkers function in Seurat was used to determine marker genes between clusters. Cell clusters were annotated manually using known canonical markers determined by FindMarkers. Cell demographics were calculated as cell counts per cluster per individual and normalized to the total cell counts per sample.

### Subsetting B cells

To further investigate B cell subpopulations, B cells were subsetted into a separate Seurat object and reanalyzed in the Seurat pipeline as above; however, the cell cycle score was not regressed out of the matrix. Cells expressing the T cell marker Cd3d were filtered out of the B cell object before further analysis.

### Pseudobulk analysis

To account for replicate samples within each condition, a pseudobulk approach was used to investigate differentially expressed genes between conditions (Thurman et al, 2021). Raw pseudobulk counts were extracted from the Seurat object for each cluster, and pseudobulk matrices were used to generate normalized counts with DEseq2 (version 1.44.0) (Love et al, 2014). Counts were filtered to remove counts ≤10 before downstream analyses. DEseq2 was used to detect differentially expressed genes between conditions [results(dds, contrast=c('condition', 'Matab', 'Naïve'), alpha = 0.05)]. The log2 fold change calculation from DEseq2 was used to generate a rank list of genes for gene set enrichment analysis (GSEA, (Subramanian et al, 2005)) using clusterProfiler (version 4.12.16) (Yu et al, 2012) to run the GSEA (method = fgsea) (Korotkevich et al, 2021) after filtering out genes with low coverage. For example, the rank list for each cluster was filtered to retain the top quartiles of genes based on the median normalized counts across all samples. The Hallmark, C2:CP, and C5 catalogs from the MsigDB database (version 7.5.1) (Subramanian et al, 2005) were used for enrichment tests.

### Histological and immunofluorescence analysis

Tissues were paraffin-embedded and sectioned for mounting on slides. Staining for hematoxylin and eosin (H&E) was performed on MLNs. Immunofluorescence slides were rehydrated in xylene, ethanol, and PBS, then antigen retrieval was performed by heating with citrate buffer. Tissue sections were permeabilized with 2% Triton X-100 in PBS, then washed three times with PBS-T prior to blocking with 5% BSA in PBS for 30 min. Slides were washed three times in PBS-T, then fluorescently-conjugated antibodies (Ki-67-FITC and B220-PE) or Hoechst, in antibody dilution buffer (2% BSA in PBS) were incubated overnight at 4 °C. Slides were washed three times with PBS-T, then blocked with Sudan black solution (0.05% Sudan black in 100% ethanol) to reduce autofluorescence. Slides were finally washed three times with PBS-T before mounting medium was used to apply a cover slip. Once dry, slides were imaged using Cytation 7 (Biotek Aligent).

### Statistics

Sample sizes were determined by natural litter sizes, which averaged 6 to 8 pups per litter. While formal power calculations were not feasible, group sizes were consistent with previous studies using this model and were sufficient to detect biologically meaningful differences in immune responses. When longitudinal pup samples were pooled within litters for a sampling point, a minimum of two litters were included per treatment group to account for the confounding effect of intra-litter correlation. Statistical analysis was performed using GraphPad Prism (version 10.3.1) and R (version 4.4.1). Immune response outcomes for the two groups were analyzed by paired or unpaired two-tailed *t*-tests. Dependent outcomes reported for three or more groups were analyzed by one-way ANOVA, and pair-wise comparisons were reported with Tukey's adjustment for multiple comparisons. Outcomes reported over time were analyzed by two-way ANOVA with repeated measures with Bonferroni's correction for multiple comparisons for two groups and Tukey's adjustment for multiple comparisons for three groups. Statistical differences were considered significant at *p* values ≤ 0.05 for all comparisons. Error bars indicate the standard error of the mean. Data were not formally assessed for normality; however, data were excluded from analysis if they were ± 2 SD from the sample population mean. Attrition of longitudinal data (less than 5%) due to low sample volume at collection or premature loss of pups or adult mice.

### Study approval

All animal studies conducted at Cornell University were approved by the Cornell University Institutional Animal Care and Use Committee (IACUC), Protocol 2022-0152. All animal studies conducted at the University of Cambridge UBS Anne McLaren facility were performed in accordance with UK Home Office guidelines and were approved by the University of Cambridge Animal Welfare and Ethical Review Board (PPL number PP6782732).

## Data availability

Source data underlying the data summarized in figures are accessible alongside respective figures. Single-cell RNA-seq data have been deposited at GEO (GSE291166) and are publicly available at the time of publication. Any additional information required to reanalyse the data reported in this paper is available from the corresponding author upon request.

The source data of this paper are collected in the following database record: biostudies:S-SCDT-10_1038-S44318-025-00582-2.

## Peer review information

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

## Acknowledgements

We thank Jeffery Ravetch for the gift of FcγRIIB KO mice, and James Crowe for the gift of rVP6. We also thank Gordon Dougan, Sallie Permar, and Deb Fowell for helpful discussions, and John Parker for manuscript review. We thank the BRC Genomics Facility and the Center for Animal Resources and Education (CARE) teams at Cornell, and the Anne McLaren Building team at the University of Cambridge for providing support with animal studies. Funding for this research was provided by the Wellcome Trust Clinical Research Career Development Fellowship to SLC (211138/A/18/Z) and the Baker Institute for Animal Health, Cornell University. The synopsis image was created with BioRender.

## Author contributions

**Tawny L Chandler**: Data curation; Software; Formal analysis; Investigation; Visualization; Methodology; Writing—original draft; Writing—review and editing. **Sarah Woodyear**: Data curation; Formal analysis; Investigation; Methodology; Writing—original draft; Writing—review and editing. **Valerie Chen**: Investigation. **Tom M Lonergan**: Investigation. **Natalie Baker**:

Investigation. **Katherine Harcourt**: Investigation. **Simon Clare**: Investigation. **Faraz Ahmed**: Investigation. **Sarah L Caddy**: Conceptualization; Resources; Data curation; Software; Formal analysis; Supervision; Funding acquisition; Validation; Investigation; Visualization; Methodology; Writing—original draft; Project administration; Writing—review and editing.

Source data underlying figure panels in this paper may have individual authorship assigned. Where available, figure panel/source data authorship is listed in the following database record: biostudies:S-SCDT-10_1038-S44318-025-00582-2.

## Disclosure and competing interests statement

The authors confirm that there are no conflicts of interest. The funding sources had no involvement in the study's design, data collection, analysis, interpretation, manuscript preparation, or the decision to publish the findings.

# Expanded View Figures

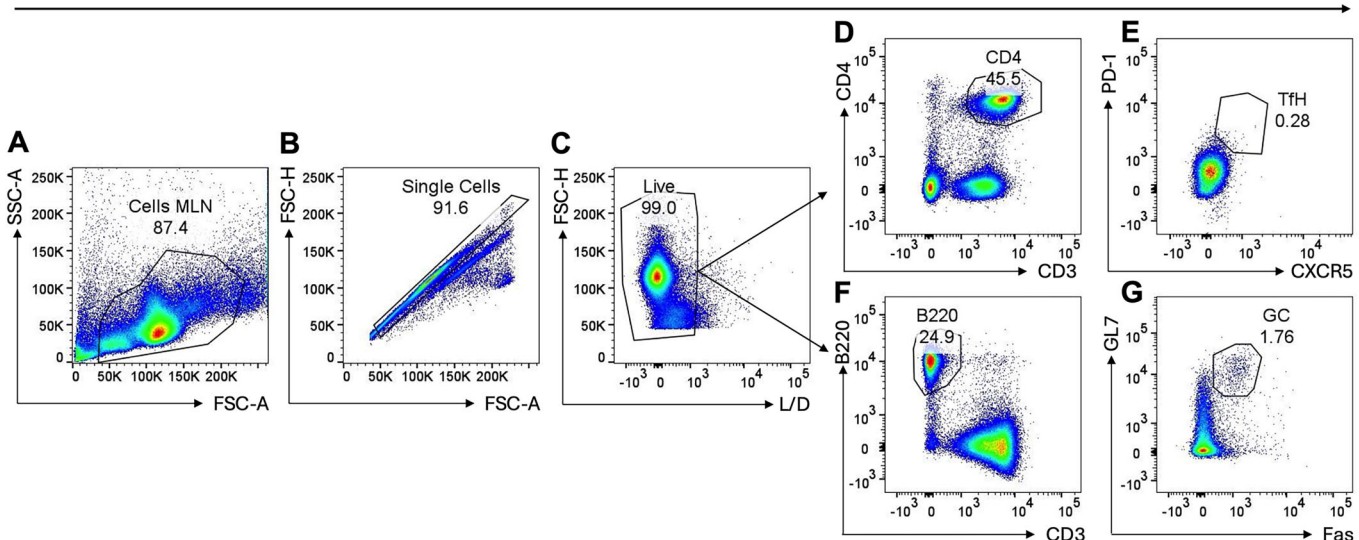

**Figure EV1. Gating strategy for identification of Tfh cells and GC B cells by flow cytometry.**

(**A**) Plot for forward versus side scatter, and mesenteric lymph node (MLN) cells gate. (**B**) Plot of forward scatter area as a function of forward scatter height within the MLN cells gate, and resulting single cells gate. (**C**) Viability dye (L/D) exclusion plot within the singlet gate, and resulting live cell gate. (**D**) Gating of CD4 cells, then gating of T follicular helper (Tfh) cells. (**E**) identified by PD-1 and CXCR5 within the CD4 gate. This panel is presented as a representative mouse in Fig. 2D. (**F, G**) Gating of B220+ B cells (**F**), then gating of germinal center (GC) B cells (**G**), identified by Fas and GL7 cells within the B220+ gate.

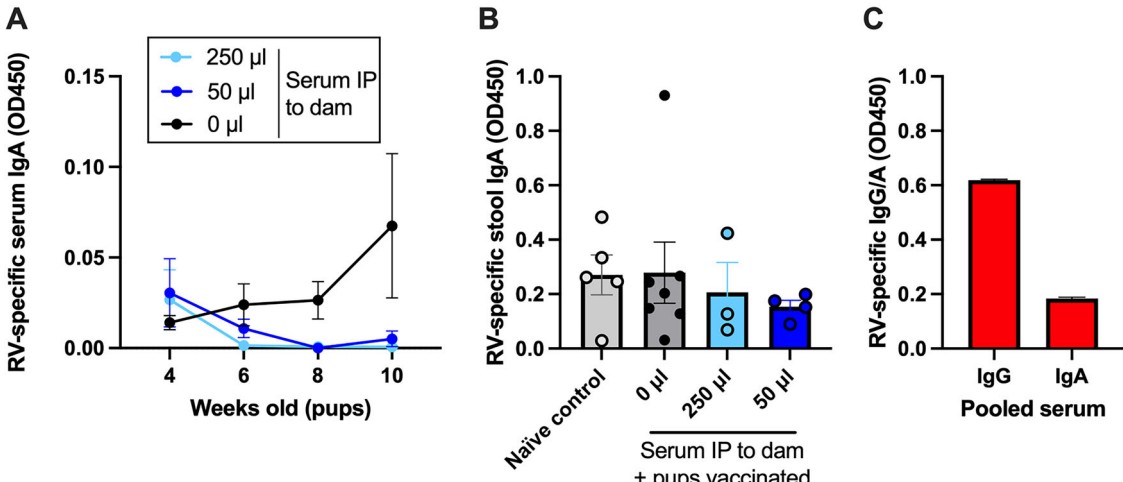

**Figure EV2. Quantification of IgA in mouse stool and serum samples in the presence or absence of MatAbs.**

(A) Longitudinal serum IgA titers following vaccination at 7 days old with different volumes of rotavirus-seropositive serum passively transferred to dams; $n = 3$, $n = 4$, and $n = 9$ for 250, 50, and 0 μL serum, respectively. (B) Stool IgA titers in adult mice from experiment (A) 14 days after challenge with rotavirus; $n = 5$, $n = 7$, $n = 4$, and $n = 3$ for naïve control, and 0, 50, and 250 μL, respectively. (C) Rotavirus-specific IgG and IgA ELISA of pooled serum used for passive transfer experiments, mean of $n = 2$ technical replicates shown. Data information: error bars indicate SEM. Source data are available online for this figure.

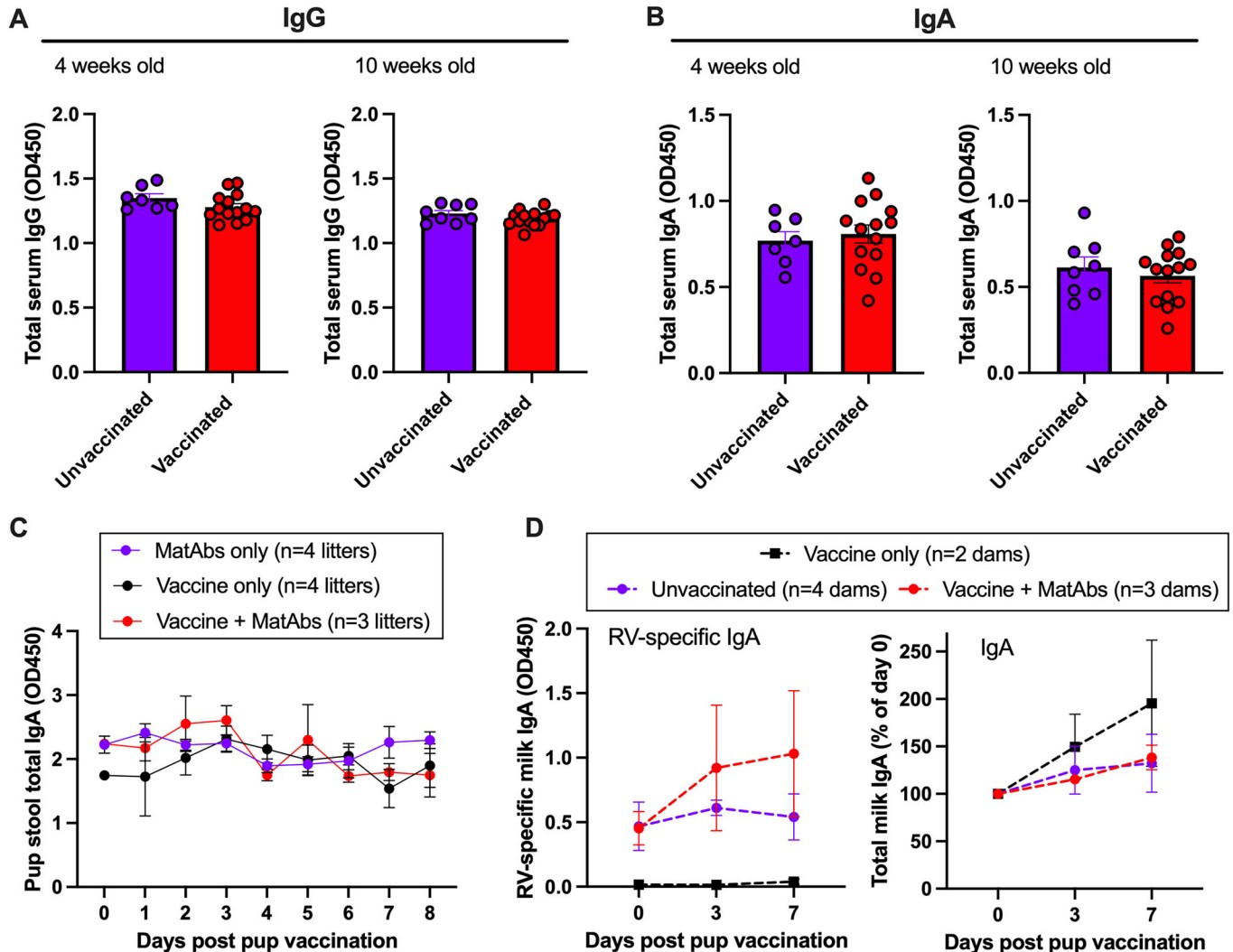

**Figure EV3. Total IgG and IgA in serum, stool, and milk samples.**

(A) Total IgG quantified by ELISA in serum samples from 4-week- and 10-week-old pups with MatAbs +/– rotavirus vaccination; $n = 7$ and $n = 14$ in unvaccinated and vaccinated pups at 4 wk and $n = 8$ and $n = 14$ in unvaccinated and vaccinated pups at 10 wk. (B) Total IgA quantified by ELISA in the same serum samples as in (A); $n = 7$ and $n = 14$ in unvaccinated and vaccinated pups at 4 wk and $n = 8$ and $n = 14$ in unvaccinated and vaccinated pups at 10 wk. (C) Total IgA ELISA in stool samples collected from pups 0 to 8 days after rotavirus vaccination. (D) Rotavirus-specific IgA and total IgA were quantified by ELISA in milk samples collected from dams at 0, 3, and 7 days after pup vaccination. Data information: error bars indicate SEM. Source data are available online for this figure.

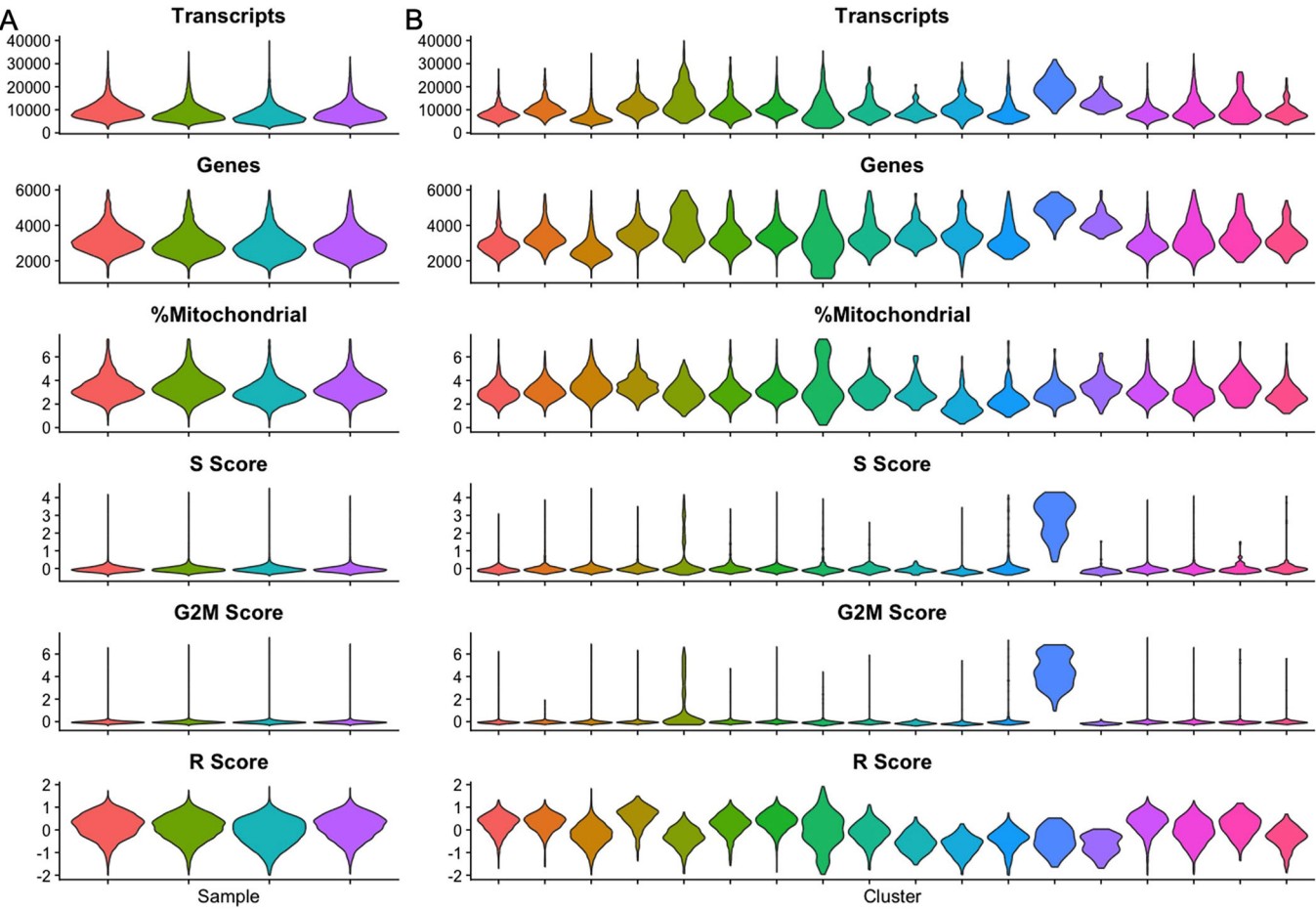

**Figure EV4. Single-cell transcriptomics of the draining mesenteric lymph node.**

(A) Violin plot showing quality control metrics (from top to bottom: transcripts per cell, genes per cell, percent mitochondrial reads per cell, S score, G2M score, and R score) for each sample (*n* = 4, x-axis) processed on the 10x Genomics Chromium instrument. (B) Violin plot showing quality control metrics for each cell type (x-axis) from MLN from pups vaccinated in the absence (*n* = 2) or presence (*n* = 2) of MatAbs.

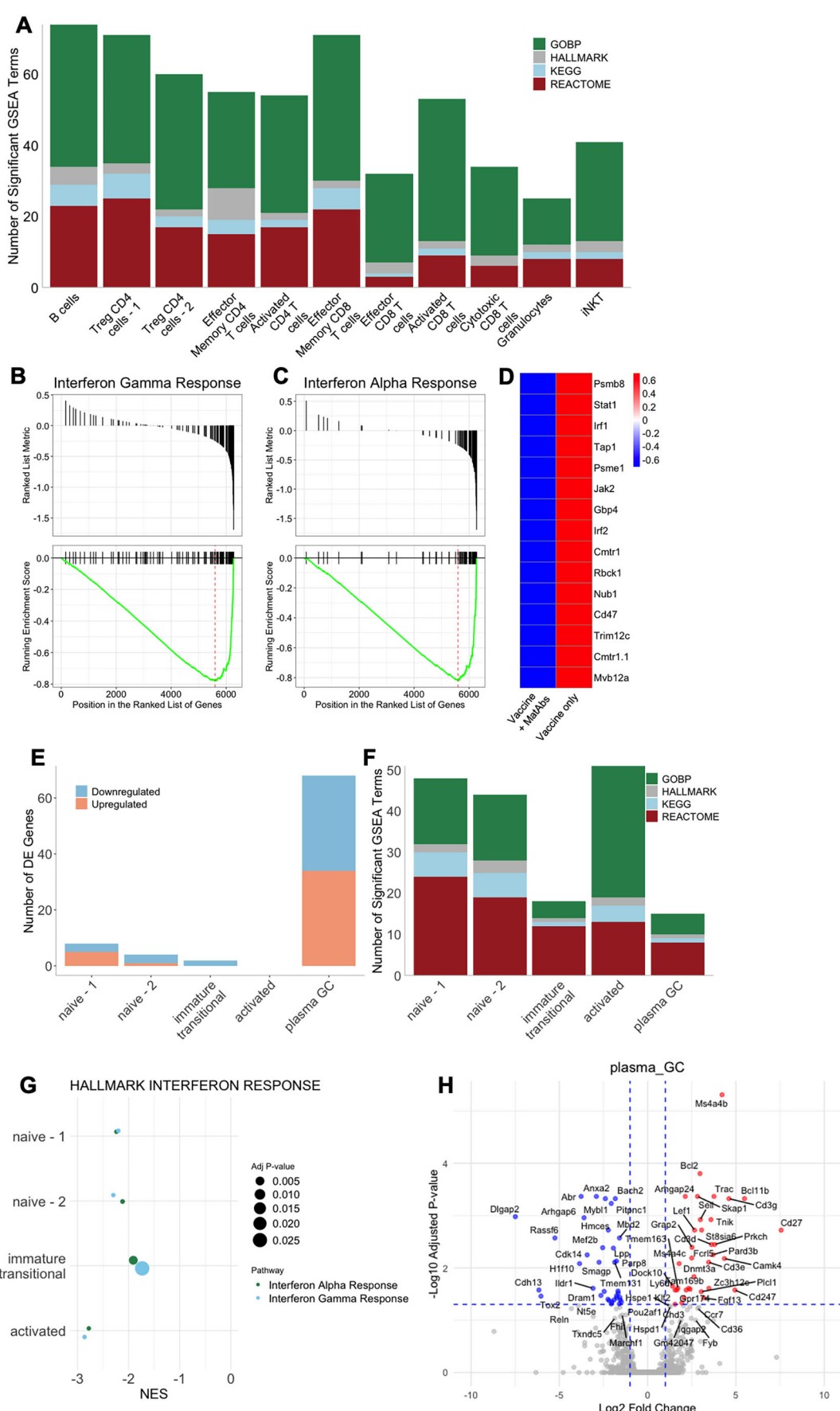

**Figure EV5. Differential gene expression in the draining mesenteric lymph node.**

(A) Counts of the strongest significantly enriched gene sets with an absolute normalized enrichment score (NES) >2 across the largest cell clusters. (B) GSEA plot of hallmark interferon gamma response in activated CD4 T cells. (C) GSEA plot of hallmark interferon alpha response in activated CD4 T cells. (D) Heatmap of top ten core enrichment genes taken from the gene set Hallmark Interferon Gamma and Hallmark Interferon Alpha Response between vaccine + MatAb and vaccine only; expression values row-normalized; unique genes from both pathways plotted. (E) Counts of differentially expressed (DE) genes (y-axis) per B cell cluster, comparing vaccine + MatAb to vaccine only with absolute log2 fold change >1 and adjusted $p$-value ≤0.05. (F) Counts of the strongest significantly enriched gene sets with an absolute normalized enrichment score (NES) >2 across B cell clusters. (G) GSEA results showing downregulation of hallmark interferon alpha response and hallmark interferon gamma response in B cell clusters. Dots are sized to denote significance; the x-axis indicates NES. (H) Volcano plot for differentially expressed genes in vaccine + MatAb ($n = 2$) compared to vaccine only ($n = 2$) (MatAb/naïve) in plasma and GC cells. Up (red) and downregulated (blue) genes with absolute log2 fold change >1 and adjusted $p$ value >0.05 shown; $-\log10$ adjusted $p$-value shown. Data information: The Wald (default in DESeq2, version 1.44.0) test was used to generate $p$-values before Benjamini–Hochberg correction for multiple testing to generate adjusted $p$-values (H).

