## [Peer Review File · The EMBO Journal]

Mechanisms of maternal antibody interference with rotavirus vaccination

Tawny Chandler, Sarah Woodyear, Valerie Chen, Tom Lonergan, Natalie Baker, Katherine Harcourt, Simon Clare, Faraz Ahmed, and Sarah Caddy

Corresponding author(s): Sarah Caddy (sarahcaddy@cornell.edu)

Review Timeline:

Submission Date:	23rd Mar 25
Editorial Decision:	2nd May 25
Revision Received:	9th Jul 25
Editorial Decision:	11th Aug 25
Revision Received:	20th Aug 25
Accepted:	25th Aug 25

Editor: Ioannis Papaioannou

Transaction Report:

Dear Sarah,

Thank you again for submitting your manuscript EMBOJ-2025-120856 for consideration by The EMBO Journal, and for your patience during peer review. It has now been seen by three experts in the field, and we have received the full set of their very detailed and well-informed reports, which you can find below.

I am very pleased to say that, as you will see, the referees all indicate interest in the study and the findings, point out the high quality of the work and the manuscript, and explain that this work makes a significant contribution to the field of rotavirus vaccination and even contributes, beyond that, to a better understanding of maternal antibody interference in oral vaccine responses. They also identify a number of limitations that should be addressed in a revised version of the manuscript. Most of them are rather minor and could be addressed by textual revision, while some call for additional experimental work that, nevertheless, appears reasonable for a single revision round and largely realistic.

Given the referees' supportive comments and positive recommendations, I would like to invite you to submit a revised version of your manuscript taking the referees' suggestions on board, along with a detailed point-by-point response addressing all referees' comments. I should add that it is The EMBO Journal policy to allow only a single round of major revision, and acceptance of your manuscript will therefore depend on the completeness of your responses in this revised version. Please let me know if you have any questions or comments that you would like to discuss with me. If there are any major points you do not agree with or cannot address during your revision, I would encourage you to share them with me as early as possible to discuss how to proceed further in the most efficient way.

We generally allow three months as standard revision time (August 1, 2025). As a matter of policy, competing manuscripts published during this period will not negatively impact our assessment of the conceptual advance presented by your study. However, we request that you contact us as soon as possible upon publication of any related work, to discuss how to proceed. Should you foresee a problem in meeting this three-month deadline, please let us know in advance and we will be able to grant an extension.

Thank you for the opportunity to consider your work for publication in The EMBO Journal. I look forward to your revision.

Best wishes,

Ioannis

Instructions for preparing your revised manuscript

1. When you are ready to submit the revision, please upload:

- A Word file of the manuscript text (including legends of main Figures, EV Figures and Tables). Please make sure that changes are highlighted (or "tracked") to be clearly visible.

- Individual production-quality figure files (one file per figure). When assembling your figures, please refer to our figure preparation guidelines in order to ensure proper formatting and readability in print as well as on screen:

If the data shown in a figure are obtained from n {less than or equal to} 2, please use scatter plots showing the individual data points.

- i. the name of the statistical test used to generate error bars and P values
- ii. the number (n) of independent experiments (please specify technical or biological replicates) underlying each data point (discussion of statistical methodology can be reported in the Materials and Methods section, but figure legends should contain a basic description of n , P, and the test applied)
- iii. the nature of the bars and error bars (s.d., s.e.m.).

- A point-by-point response to the referees' comments, with a detailed description of the changes made (as a word file). All referees' concerns must be fully addressed and their suggestions taken on board. When preparing your letter of response to the referees' comments, please bear in mind that this will form part of the Review Process File and will therefore be available online to the community. Please note that you have the possibility to opt out of the transparent process at any stage prior to publication by letting the editorial office know (contact@embojournal.org); if you do opt out, the Review Process File link will point to the following statement: "No Review Process File is available with this article, as the authors have chosen not to make the review process public in this case.". For more details on our Transparent Editorial Process, please visit our website: <https://www.embopress.org/page/journal/14602075/authorguide#transparentprocess>

- Expanded View (EV) files (replacing Supplementary Information) that are collapsible/expandable online. A maximum of 5 EV Figures can be typeset. EV Figures should be cited as "Figure EV1, Figure EV2" etc. in the text, and their respective legends should be included in the manuscript file after the legends of regular figures. See detailed instructions regarding Expanded View files here: <https://www.embopress.org/page/journal/14602075/authorguide#expandedview>

- For the figures that you do NOT wish to display as Expanded View figures, they should be bundled together with their legends in a single PDF file called "Appendix", which should start with a short Table of Contents (including page numbers). Appendix figures should be referred to in the main text as: "Appendix Figure S1, Appendix Figure S2" etc. Please see detailed instructions here: <https://www.embopress.org/page/journal/14602075/authorguide#expandedview>

- A complete author checklist, which you can download from our author guidelines (<https://www.embopress.org/page/journal/14602075/authorguide>). Please note that the checklist will also be part of the Review Process File.

2. Please note that no statistics should be calculated and shown in Figures if $n=2$. Please also note that each p value should be reported as an exact value.

3. Before submitting your revision, primary datasets (and computer code, where appropriate) produced in this study need to be deposited in appropriate public databases (see <https://www.embopress.org/page/journal/14602075/authorguide#dataavailability>). In particular, we kindly request you to deposit all sequencing data in an appropriate repository. The accession numbers, database, and the specific URLs (links) should be listed in a formal "Data availability" section (placed after Methods), following the example below:

"The RNA-seq datasets produced in this study are available in the following database:
Gene Expression Omnibus GSE46843 (<https://www.ncbi.nlm.nih.gov/geo/query/acc.cgi?acc=GSE46843>)"

*** All links should resolve to a page where the data can be accessed. ***

*** Please remember to provide in the Data availability section of your revised manuscript reviewer passwords if the datasets are not yet public. ***

*** The Data Availability Section is restricted to new primary data that are part of this study. In case you have no data that require deposition in a public database, please state so instead of referring to the database: "Our study includes no data deposited in public repositories." under the heading "Data availability". ***

4. The materials and methods need to be described in the manuscript using our structured methods format, which is now required for all research articles. According to this format, the Methods section includes a single "Reagents and Tools Table" - listing key reagents, experimental models, software and relevant equipment including their sources and relevant identifiers - followed by a "Methods and Protocols" section describing the methods. Please download and fill our Reagents and Tools Table template (.docx), which you can find in our author guide:

<https://www.embopress.org/page/journal/14602075/authorguide#structuredmethods>. When submitting your revised manuscript, please do not include the Reagents and Tools Table in the Methods section of the manuscript but instead upload it as a separate file choosing the file type "Reagent Table".

5. Please check that the title and the abstract of the manuscript are brief, yet explicit, even to non-specialists. The length of the title should not exceed 100 characters, and the abstract should be a single paragraph not exceeding 175 words.

6. Please also note our reference format: <https://www.embopress.org/page/journal/14602075/authorguide#referencesformat>.

8. Please remember: digital image enhancement is acceptable practice, as long as it accurately represents the original data and

conforms to community standards. If a figure has been subjected to significant electronic manipulation, this must be noted in the figure legend or in the "Materials and Methods" section. The editors reserve the right to request original versions of figures and the original images that were used to assemble the figure.

9. Our journal encourages inclusion of data citations in the reference list to directly cite datasets that were obtained from public databases. Data citations in the article text are distinct from normal bibliographical citations and should directly link to the database records from which the data can be accessed. In the main text, data citations are formatted as follows: "Data ref: Smith et al, 2001" or "Data ref: NCBI Sequence Read Archive PRJNA342805, 2017". In the Reference list, data citations must be labeled with "[DATASET]". A data reference must provide the database name, accession number/identifiers, and a resolvable link to the landing page from which the data can be accessed at the end of the reference. Further instructions are available at: <https://www.embopress.org/page/journal/14602075/authorguide#referencesformat>.

10. We request authors to consider both actual and perceived competing interests. Please review our policy (<https://www.embopress.org/page/journal/14602075/authorguide#conflictofinterest>) and update your competing interests statement if necessary. Please name this section 'Disclosure and competing interests statement' and place it after the Acknowledgements section.

11. Please note that all corresponding authors are required to provide an ORCID ID upon submission of a revised manuscript (<https://orcid.org/>). Please find instructions on how to link your ORCID ID to your account in our manuscript tracking system in our Author guidelines (<https://www.embopress.org/page/journal/14602075/authorguide#authorshipguidelines>).

12. We use CRediT to specify the contributions of each author in the journal submission system. CRediT replaces the author contribution section, which should be removed from the manuscript. Please use the free text box to provide more detailed descriptions. See also guide to authors: <https://www.embopress.org/page/journal/14602075/authorguide#authorshipguidelines>.

14. We would also welcome the submission of cover suggestions or motifs to be used by our Graphics Illustrator in designing a cover.

15. Please use the link below to submit your revision:
<https://emboj.msubmit.net/cgi-bin/main.plex>

Referee #1:

The authors investigate mechanisms of transplacental antibody transfer in the context of rotavirus in order to inform vaccination strategies for infants. They found that maternal antibody interference is a critical mechanism limiting neonatal B cell responses, leaving them vulnerable to infection. Their data show a limited role for FcyRIIB, with clearance of vaccine by maternal antibodies as the primary driver of reduced neonatal immunity. The experimental strategies used to uncover the mechanistic contributions and translational methods to interrogate these differences was very thorough. This was a very important finding with public health implications.

Minor Issues:

1. Authors interpret the role of IgA to be the same as IgG in maternal and neonatal immunity. IgA is not transferred across the placenta and only binds FcyRIIB with a very low affinity in limited circumstances, which likely is the explanation for their data.
2. For Figure 6A, please clarify. The image indicates that pups were infected at 7 days old, however the text specifies that this was oral vaccination.
3. When addressing differences between murine and human pregnancies and antibody transfer, it is important to acknowledge that mice are born developmentally immature and behind human babies at term delivery. This is an important consideration especially considering your vaccination timepoint is still before birth in human equivalent terms developmentally.
 - a. Veru, F., Laplante, D. P., Luheshi, G., & King, S. (2014). Prenatal maternal stress exposure and immune function in the offspring. *Stress*, 17(2), 133-148. <https://doi.org/10.3109/10253890.2013.876404>

Referee #2:

In Chandler et al., the authors developed a murine rotavirus model of maternal antibody (matAb) interference to rotavirus-specific immune responses in the suckling neonate. They find that matAbs, even at low levels, block replication of an attenuated murine rotavirus in neonatal pups that results in limited formation of germinal centers compared to pups receiving no matAbs. They show that FcγRIIB signaling did not impact IgG matAb interference and that matAb interference can be overcome by giving a higher dose of the attenuated murine rotavirus. I very much enjoyed reading this well written manuscript. It presents interesting data that significantly advances the field of maternal/neonatal rotavirus immunity.

The majority of suggestions I have are relatively minor (see below), however the presentation and interpretation of Figure 5 has flaws that need to be addressed. I'll address these first:

-Lines 236-240 could go to the discussion (generally, the results section can be condensed, with select text going to the discussion)

-Immediately after lines 250-253 (referencing Figure 5B/C), the authors state 'We propose this data supports the hypothesis that vaccine clearance is a key mechanism by which interference occurs; when MatAbs bind to vaccine particles in the intestine this results in clearance of MatAb-vaccine complex. This process can be described as 'antigen consumption' (Tas et al, 2022), and in turn this alters the concentration gradient between MatAbs in the circulation and MatAbs in the intestinal mucosa such that more MatAbs are drawn out of circulation.' However Figures 5B/C do not support this hypothesis as RV-specific IgG levels in stool are not statistically higher in matAb+vaccine compared to the matAb group. If the authors want to claim this, and believe there is a statistically significant difference, more mice may be required. All the authors should say is that serum RV-specific IgG antibodies were lower in vaccinated compared to unvaccinated pups and then leave the speculation about matAb-complexes to the discussion. My justification for this is that the authors didn't measure matAb-vaccine complexes (even though they state they did 'Next, we questioned what the fates of rotavirus-specific IgG MatAbs were once they had bound to their target'). To do this, the authors would need to isolate RV-specific antibodies from stool and then take those antibodies and run qPCR to quantify the amount of RV bound to the matAbs.

-Were there differences in RV-specific IgA in vaccinated+unvaccinated pup serum in the Figure 5 experiments? Please include. -It is possible that RV-specific IgA antibodies are higher in the stool of matAb+vaccine pups compared to matAb pups because the attenuated murine rotavirus infection results in replication and production of infectious virus in the pups' oral cavity (as previously shown for wild-type murine rotavirus by Ghosh et al., 2020 PMID: 35768512). This would result in inoculation of the mammary gland with pup oral cavity-derived rotavirus, resulting in increased IgA in milk (also previously demonstrated by Ghosh et al). This would also explain the increased IgA in pup stool. To test this, the authors should measure RV-specific and total IgA (and IgG) antibodies in milk of the dams as well as rotavirus viral RNA in milk via qPCR. This will help explain the data in Figure 5.

-274-276 should be moved to discussion. Also the authors are able to easily test this hypothesis (that their results are due to rotavirus induced IFN-γ leading to increased pIgR expression and transfer of IgA across the gut) by measuring total IgA responses in stool. Please include this measurement to better understand Figure 5 data.

-In line 251 the authors state 'we also observed a surprising reduction in total circulating IgG MatAbs in vaccinated pups compared to unvaccinated controls (Fig 5B/C)' however Figure 5B/C showed rotavirus-specific IgG not total IgG. Did the authors mean RV-specific IgG? To better understand the data presented in Figure 5B,C,D,E,&F, the authors should also measure total IgG and IgA in serum.

-To aid in the reader's understanding of Figure 5D/E the authors could note that there should be no RV specific antibodies in the stool of 9-day-old pups that were vaccinated at 7 days of age because there wasn't enough time to generate a RV specific immune response. Therefore, what is being observed is MatAbs only.

-Generally, I think the authors should deemphasize the mention of 'MatAb-vaccine/virus complexes' as they do not measure this. They can state that there was reduced replication that was likely mediated by matAb binding and clearance, but matAb-virus complexes were not measured.

Single cell data: In general this results section feels very long. I would recommend condensing, with a focus on your most significant/impactful data and possibly shifting one of these two figures to supplementary.

Other comments:

Line 174: please state here the different doses of rotavirus used

In Figure 1B, can you denote when the pups were vaccinated *7 weeks

Figure 3D is really interesting. Your dam serum RV-specific IgA data would suggest that either (1) RV-specific IgA in pooled serum that was IP transferred is low/non-existent or (2) the RV-specific IgA in pooled serum that was IP transferred is quickly shuttled to the mammary gland/milk. Could you add in the main text or supplementary figure the levels of RV-specific IgG and IgA in the pooled serum from RV-positive mice that was IP transferred? This would help confirm that there was RV-specific IgA in the pooled serum (similar to what you show for individual mice in Figure 1D). Assuming there was RV-specific IgA in the pooled serum, then the reader could interpret Figure 1D as serum RV-specific IgA quickly trafficking to the mammary gland. This is quite fascinating as dogma in the breast milk IgA field is that IgA is mostly derived from local IgA-secreting plasma cells in the mammary gland. But your data suggests that serum IgA could also contribute.

Your Figure 4C is quite fascinating (as you mention, this is evidence of IgA-specific interference mechanisms). Could you include RV-specific IgA in serum from these experiments? It will help better understand the RV-specific IgA results in stool.

Figure 1 D/C could you put '4 weeks old' at the top of those graphs (or otherwise denote on the graph what time point that is)

Line 433: Could you add a comment whether Yang et al, 2019; Johansson et al, 2008; Zhou et al, 2022, like your work, used a lower/attenuated dose of RV to explain the differences between these works and the Muleta preprint?

Line 466: I would use a different word than 'shedding' here. Shedding usually refers to viral RNA/particles/infectious virus - a more appropriate word for antibodies would be 'transfer'.

Line 468: Can easily test this by measuring total IgA in stool (as mentioned above)

Line 493-494: Can you provide a citation for this?

Line 512: I would use a different word than blighted. Blighted seems too strong as despite the side effects, the world still has two effective rotavirus vaccines that are used in pediatric vaccination schedules.

Line 576-577, 590: Could you please mention the source of your rotavirus-specific polyclonal antibodies (sheep and rabbit) in the manuscript?

Referee #3:

EMBOJ 120856

Title: Mechanisms of maternal antibody interference to rotavirus vaccination

1-General summary and opinion about the principal significance of the study, its questions, and findings

The primary aim of this study was to elucidate the mechanisms by which maternal antibodies (MatAbs) interfere with the efficacy of live-attenuated rotavirus vaccines in human infants. The authors successfully established and standardized a neonatal mouse model to mimic MatAb interference, representing a significant methodological advancement. This innovative model allowed the authors to explore the suppression of active IgG and IgA antibody responses in serum and feces and the reduction in B and T cell germinal center formation within the mesenteric lymph nodes (MLN). Additionally, the study demonstrated that MatAbs significantly impair vaccine live-attenuated virus replication in the gut. Importantly, the authors showed that FcγRIIB signaling was not essential for MatAb-mediated interference. The study is also the first to report single-cell analysis of the lymph node response to rotavirus infection, providing valuable insight into the anti-viral pathways stimulated.

Overall, the study provides compelling evidence that the primary mechanism of interference involves MatAb-mediated clearance of the vaccine virus before it can effectively engage the neonatal immune system. This work is a well-designed and carefully executed study that offers important insights into rotavirus immunology. Its findings are highly relevant not only to the field of rotavirus vaccination but also to our broader understanding of maternal antibody interference in oral vaccine responses.

2- Specific major concerns essential to be addressed to support the conclusions

2.1- A major concern is the lack of discussion and consideration of other components of maternal immunity beyond antibody transfer. Recent studies have highlighted that, in addition to maternal antibodies, immune cells and cytokines are also transferred from mother to offspring - both transplacentally and via breastfeeding - and can play a significant role in shaping neonatal immune responses. This aspect is not addressed in your study. Please justify why these components were not included in your so deep analysis and discuss how their omission might affect the interpretation of your findings, particularly in the context of early-life vaccination strategies.

2.2- Another major concern is the absence of virus neutralization assays in your evaluation of the antibody response. While ELISA can provide information on the presence of antigen-specific binding antibodies (IgM, IgA and IgG), it does not indicate whether these antibodies are functionally capable of neutralizing the virus. Given that protection is more closely associated with neutralizing antibody titers, this omission limits the interpretation of vaccine efficacy. Additionally, ELISA results are presented as optical density (OD) values at a single sample dilution, which can be misleading and may not reflect the actual magnitude of the response. It is standard practice to report ELISA titers as the reciprocal of the highest dilution yielding a positive signal. Please consider including these data or discussing these limitations more explicitly.

Lines 38-39:

In the Introduction, it is important to explain the species-specific expression of the neonatal Fc receptor (FcRn) in the placenta. For example, horses and cattle do not express FcRn in the placenta, and therefore, there is no transplacental transfer of maternal antibodies in these species. In contrast, mice express FcRn both in the placenta and in the neonatal intestine, allowing maternal IgG transfer during gestation and also postnatally through colostrum and milk. Please clarify how this mechanism operates in your model vs humans. Is FcRn expressed in the human placenta and neonatal gut? Justify why the mouse is considered an appropriate model for studying maternal antibody transfer in humans, and discuss the limitations of using this model and not a primate model.

Line 95:

What was the rationale for choosing day 7 postnatal as the time point for vaccination of pups? You mention this is meant to mimic human infant vaccination at 8 weeks of age - please explain how this timing aligns developmentally and immunologically between mice and humans.

More importantly, have you thought in testing for the presence of IgM, IgA and IgG antibody-secreting cells in the gut and in the spleen? You need to check for the activated cells that could be primed even in the presence of very high titers of maternal IgG. In calf models of RVA it was reported that IgM switching to IgA and mostly IgG1 was impaired by colostrum passive antibodies, but most importantly the active immune response was modulated due to the cytokines of the colostrum not only the antibodies, what can you say about that in your model?

Lines 101-108:

You evaluated IgG and IgA antibody responses in serum and feces. However, no data are shown for IgM responses. Could you explain why IgM was not assessed, considering it is the first isotype produced during a primary immune response and could provide additional insights into vaccine immunogenicity and interference due to maternal antibodies

Lines 397-407

Another important point to address is the rationale behind the assumption that parenteral vaccination with a double-layered particle (DLP) vaccine would be more effective in overcoming the interference of maternal antibodies than oral administration of a live attenuated virus. Given that live oral vaccines can replicate at mucosal surfaces and potentially induce local immunity despite the presence of maternal antibodies, it is not immediately clear why the parenteral route using a non-replicating antigen would be expected to perform better in this context. Please justify this choice and clarify whether any comparative data or previous studies support this hypothesis.

Furthermore, while you observed an increase in ELISA signal following parenteral DLP vaccination in the presence of maternal antibodies, it is important to consider that maternal serum IgG - which can be systemically absorbed in mice - may also neutralize the DLP vaccine, potentially impacting its effectiveness. An increase in ELISA-detected binding antibodies does not necessarily correlate with protective immunity. Did you assess seroconversion in terms of neutralizing antibody titers, which are more directly associated with protection? Including this information would greatly strengthen your conclusions regarding vaccine efficacy in the context of maternal antibody interference.

3- Minor concerns that should be addressed

Section in line 543

In M&M please provide the virus titer in FFU/ml also and compared with the dose of the vaccines used in humans. It will be also useful to estimate the ug of DLP administered in the parenteral vaccine, as we know that at least 10E6-10E7 TCID50/dose are needed to see a satisfactory response with inactivated whole virus.

Section in line 575. Why you did not tested the Ab titer in serial four-fold dilutions?
Why IgM Abs are missing?

Section in line 607. Why you did not do ELISPOT of ASC in the cell obtained from MLN, or other immune tissues?

In the figure 1. (F) and figure 3 (E) and figure 4 (C) it is stool IgG instead of serum IgG?

4- Any additional non-essential suggestions for improving the study
(which will be at the author's/editor's discretion)

I think it is an excellent research, congratulation to the author for the hard work!

Many thanks to the reviewers for their comments. We are grateful for this careful consideration of our work and have responded to each reviewer comment in turn as presented below.

Referee #1:

The authors investigate mechanisms of transplacental antibody transfer in the context of rotavirus in order to inform vaccination strategies for infants. They found that maternal antibody interference is a critical mechanism limiting neonatal B cell responses, leaving them vulnerable to infection. Their data show a limited role for FcγRIIB, with clearance of vaccine by maternal antibodies as the primary driver of reduced neonatal immunity. The experimental strategies used to uncover the mechanistic contributions and translational methods to interrogate these differences was very thorough. This was a very important finding with public health implications.

Minor Issues:

1. Authors interpret the role of IgA to be the same as IgG in maternal and neonatal immunity. IgA is not transferred across the placenta and only binds FcγRIIB with a very low affinity in limited circumstances, which likely is the explanation for their data.

We agree that IgA and IgG MatAbs will play separate roles in neonatal immunity in the context of FcγRIIB. We have edited the text in the associated results section (now lines 217-218) and in the discussion (now line 464) to make it clear that we are considering IgG MatAbs only for FcγRIIB experiments. However it still remains unclear how a lack of FcγRIIB signaling when both IgG and IgA titers are low permits neonatal B cell class switching to IgG but not IgA. We have stated in our discussion (now lines 468 to 471) that we cannot yet make conclusions about the impact of IgG MatAbs relative to IgA MatAbs, but future work aims to separate the relative contributions of IgG and IgA MatAbs in interference with neonatal immune responses.

2. For Figure 6A, please clarify. The image indicates that pups were infected at 7 days old, however the text specifies that this was oral vaccination.

Thank you for pointing this out, this has been clarified in the Figure 6A. Pups were orally vaccinated at 7 days old.

3. When addressing differences between murine and human pregnancies and antibody transfer, it is important to acknowledge that mice are born developmentally immature and behind human babies at term delivery. This is an important consideration especially considering your vaccination timepoint is still before birth in human equivalent terms developmentally.

a. Veru, F., Laplante, D. P., Luheshi, G., & King, S. (2014). Prenatal maternal stress exposure and immune function in the offspring. *Stress*, 17(2), 133-148. <https://doi.org/10.3109/10253890.2013.876404>

Thank you for highlighting this issue, we agree this is a significant challenge when using mice to study human neonatal immunology. We chose a 7d timepoint for infecting pups as this is a well-recognized and widely used age for studying enteric virus infections in mice. The strain of murine rotavirus we used throughout this study (EMcN) was originally characterized at pups infected at 5 days old, and we chose to delay infection by 2 days to allow more maturation of the immune response. We have edited the text accordingly to provide more reasoning for the 7-day mouse vaccination timepoint (now lines 99 to 103, 513 to 515), and also note the World Health Organization recommends rotavirus vaccination is administered from 6-8 weeks of age.

Referee #2:

In Chandler et al., the authors developed a murine rotavirus model of maternal antibody (matAb) interference to rotavirus-specific immune responses in the suckling neonate. They find that matAbs, even at low levels, block replication of an attenuated murine rotavirus in neonatal pups that results in limited formation of germinal centers compared to pups receiving no matAbs. They show that FcγRIIB signaling did not impact IgG matAb interference and that matAb interference can be overcome by giving a higher dose of the attenuated murine rotavirus. I very much enjoyed reading this well written manuscript. It presents interesting data that significantly advances the field of maternal/neonatal rotavirus immunity.

The majority of suggestions I have are relatively minor (see below), however the presentation and

interpretation of Figure 5 has flaws that need to be addressed. I'll address these first:

-Lines 236-240 could go to the discussion (generally, the results section can be condensed, with select text going to the discussion)

We have moved this sentence to the discussion as suggested (now lines 476 to 478).

-Immediately after lines 250-253 (referencing Figure 5B/C), the authors state 'We propose this data supports the hypothesis that vaccine clearance is a key mechanism by which interference occurs; when MatAbs bind to vaccine particles in the intestine this results in clearance of MatAb-vaccine complex. This process can be described as 'antigen consumption' (Tas et al, 2022), and in turn this alters the concentration gradient between MatAbs in the circulation and MatAbs in the intestinal mucosa such that more MatAbs are drawn out of circulation.' However Figures 5B/C do not support this hypothesis as RV-specific IgG levels in stool are not statistically higher in matAb+vaccine compared to the matAb group. If the authors want to claim this, and believe there is a statistically significant difference, more mice may be required. All the authors should say is that serum RV-specific IgG antibodies were lower in vaccinated compared to unvaccinated pups and then leave the speculation about matAb-complexes to the discussion. My justification for this is that the authors didn't measure matAb-vaccine complexes (even though they state they did 'Next, we questioned what the fates of rotavirus-specific IgG MatAbs were once they had bound to their target'). To do this, the authors would need to isolate RV-specific antibodies from stool and then take those antibodies and run qPCR to quantify the amount of RV bound to the matAbs.

We agree that our interpretation of these results from Figure 5B/C are better suited to the discussion, and have moved them accordingly (now lines 481 to 499). Apologies for the typing error, it should be *antibody* consumption not antigen consumption. This has been edited in the discussion text. We also agree that we have not specifically measured MatAb-vaccine complexes, and so have changed our wording throughout this manuscript to reflect this.

In addition, we have now extended our discussion around the possible reasons for our observation that MatAbs are significantly decreased in serum after vaccination. Whilst clearance of IgG in stool remains a possibility, we agree we have not proven this experimentally which could be due to insufficient numbers of mice. Overall we have concluded the precise mechanism of IgG MatAb clearance remains uncertain and we will address this in future work. We have however made progress on the fate of IgA MatAbs from milk thanks to your helpful suggestions, and these are detailed in sections below.

-Were there differences in RV-specific IgA in vaccinated+unvaccinated pup serum in the Figure 5 experiments? Please include.

Figure 1D presents the serum RV-specific IgA quantified in the litters of pups vaccinated and unvaccinated in Figure 5 B/C (which shows RV-specific IgG). This showed that no RV-specific maternal IgA was transferred to the pup circulation in either group, and only pups vaccinated in the absence of MatAbs mounted an IgA response that was detectable in serum and stool (Figure 1E). We have modified the text to highlight this (now lines 132; 265).

-It is possible that RV-specific IgA antibodies are higher in the stool of matAb+vaccine pups compared to matAb pups because the attenuated murine rotavirus infection results in replication and production of infectious virus in the pups' oral cavity (as previously shown for wild-type murine rotavirus by Ghosh et al., 2020 PMID: 35768512). This would result in inoculation of the mammary gland with pup oral cavity-derived rotavirus, resulting in increased IgA in milk (also previously demonstrated by Ghosh et al). This would also explain the increased IgA in pup stool. To test this, the authors should measure RV-specific and total IgA (and IgG) antibodies in milk of the dams as well as rotavirus viral RNA in milk via qPCR. This will help explain the data in Figure 5.

Thank you for this valuable suggestion. We typically detect seroconversion in dams after pups are vaccinated with live attenuated rotavirus, so we agree that dam infection can occur. To address the question of whether dam infection with the vaccine strain occurs via the mammary gland or orally, we tested milk collected 7 days after pup vaccination for the presence of rotavirus RNA by qPCR. However, we were unable to detect RV RNA in milk from dams of vaccinated pups. This could either reflect the sensitivity of our assay or the sample studied; Ghosh et al quantified RV RNA from mammary tissue directly, whereas we chose not to perform terminal culls on dams in order to longitudinally sample pups up to weaning.

We have quantified total IgG and total IgA and RV-specific IgA/IgG in the milk of dams in our experimental pipeline. To achieve this, we set up new pup vaccine experiments and collected milk samples from dams on the day of pup vaccination, plus 3 and plus 7 days post vaccination. These results are now presented in EV Figure 3D and lines 289 to 295). We found that RV-specific IgA in milk showed some increase post infection, but was not significantly different between the dams with RV-specific maternal antibodies whose pups were vaccinated and those that were unvaccinated. Total IgA increased in dams whose litters were vaccinated by day 7, although this was not significantly different with this number of mice. We suggest that the general upward trend in total IgA in milk is due to an upregulation of pIgR. However, this does not explain the increase of RV-specific IgA in stool we observed as the timing is different; we observed a rapid increase in stool RV-specific IgA (peaking day 2-3, and falling by day 7), whereas milk RV-specific IgA slowly rises up to day 7. We suggest that this new data provides support for our conclusion that milk rotavirus-specific IgA is being cleared via the gastrointestinal tract.

For completeness, we also quantified RV-specific IgG in milk, but minimal RV-specific IgG was detected which is in agreement with figure 5D where minimal RV-specific IgG was detected in the stool of pups. This data has not been included.

-274-276 should be moved to discussion. Also the authors are able to easily test this hypothesis (that their results are due to rotavirus induced IFN- γ leading to increased pIgR expression and transfer of IgA across the gut) by measuring total IgA responses in stool. Please include this measurement to better understand Figure 5 data.

We have moved the sentence to discussion as recommended (now lines 496 to 499). Thank you for this excellent suggestion to test the stool samples for total IgA. We collected further samples from litters described above and were able to perform this experiment. The results are presented in Figure EV3C and described in lines 288 to 289. We also measured RV-specific IgA in these additional litters to strengthen the results presented in figure 5F. We observed that total IgA remained consistent in the pup stool samples throughout the collection period, which is in direct contrast to the rise in RV-specific IgA detected 2-3 days post vaccination. These results are in agreement with the milk IgA results described above. Overall, these additional experiments support the conclusion that MatAbs are mediating vaccine clearance to result in increased RV-specific MatAb excretion in the stool.

-In line 251 the authors state 'we also observed a surprising reduction in total circulating IgG MatAbs in vaccinated pups compared to unvaccinated controls (Fig 5B/C)' however Figure 5B/C showed rotavirus-specific IgG not total IgG. Did the authors mean RV-specific IgG? To better understand the data presented in Figure 5B,C,D,E,&F, the authors should also measure total IgG and IgA in serum.

Thank you for catching this oversight, we meant RV-specific IgG and have edited the text accordingly (now lines 261). We have now quantified total IgG and IgA in these same serum samples for which sufficient volume was still available. A single dilution of serum was used due to the volume issue, and these results are presented in Figure EV3A,B. We were able to verify with 3 samples for each group that there was clear dose-dependency for these results, and no significant differences were observed as shown by the curves presented here.

Overall our results show that there was no significant difference in total IgG or IgA at either 4 or 10 weeks of age, despite significant differences in RV-specific IgG. This provides additional support to the conclusion that RV-specific IgG MatAbs are being specifically depleted after vaccination, although the precise means are yet unknown.

-To aid in the reader's understanding of Figure 5D/E the authors could note that there should be no RV specific antibodies in the stool of 9-day-old pups that were vaccinated at 7 days of age because there wasn't enough time to generate a RV specific immune response. Therefore, what is being observed is MatAbs only. We have added a sentence to the text to explain this more clearly (now lines 275 to 277).

-Generally, I think the authors should deemphasize the mention of 'Matab-vaccine/virus complexes' as they do not measure this. They can state that there was reduced replication that was likely mediated by matAb binding and clearance, but matAb-virus complexes were not measured.

Thank you for this suggestion, we agree and have edited the text to remove mention of MatAb-vaccine complex quantification.

Single cell data: In general this results section feels very long. I would recommend condensing, with a focus on your most significant/impactful data and possibly shifting one of these two figures to supplementary.

Thank you for the comment, we have now condensed this section to focus on the most significant findings and have shifted part of the B cell figure to expandable view supplementary figures.

Other comments:

Line 174: please state here the different doses of rotavirus used

We have added a note in the text to specify this was 1 FFU and 0.1 FFU (now line 187).

In Figure 1B, can you denote when the pups were vaccinated *7 weeks

Throughout this study, all vaccines were administered to mice at 7 days of age (with the exception of Figure 8A). We have added a sentence reflecting this when Fig 1B is first referenced in the text (lines 100 to 101).

This timing is also presented in the schematic diagram in Fig 1A.

Figure 3D is really interesting. Your dam serum RV-specific IgA data would suggest that either (1) RV-specific IgA in pooled serum that was IP transferred is low/non-existent or (2) the RV-specific IgA in pooled serum that was IP transferred is quickly shuttled to the mammary gland/milk. Could you add in the main text or supplementary figure the levels of RV-specific IgG and IgA in the pooled serum from RV-positive mice that was IP transferred? This would help confirm that there was RV-specific IgA in the pooled serum (similar to what you show for individual mice in Figure 1D). Assuming there was RV-specific IgA in the pooled serum, then the reader could interpret Figure 1D as serum RV-specific IgA quickly trafficking to the mammary gland. This is quite fascinating as dogma in the breast milk IgA field is that IgA is mostly derived from local IgA-secreting plasma cells in the mammary gland. But your data suggests that serum IgA could also contribute.

As serum was pooled from mice following infection, pooled serum had similar levels of IgG and IgA as natural infection in figure 3D. The serum pool was tested for RV-specific serum IgG and IgA and this data has now been included in Fig EV2. We agree that our findings indicate IgA can be rapidly trafficked to the mammary gland following intraperitoneal injection, which is intriguing!

Your Figure 4C is quite fascinating (as you mention, this is evidence of IgA-specific interference mechanisms). Could you include RV-specific IgA in serum from these experiments? It will help better understand the RV-specific IgA results in stool.

We have now measured RV-specific IgA in the serum from the mice in these experiments, and added these results to Figure 4C. This shows that MatAbs mediate interference with both serum and stool IgA, but not serum IgG. This suggests that interference can be the level of class switching of antibody, instead of impacting mucosal delivery of IgA alone.

Figure 1 D/C could you put '4 weeks old' at the top of those graphs (or otherwise denote on the graph what time point that is)

Yes, this has been done.

Line 433: Could you add a comment whether Yang et al, 2019; Johansson et al, 2008; Zhou et al, 2022, like your work, used a lower/attenuated dose of RV to explain the differences between these works and the Muleta preprint?

We have clarified that Johansson et al and Zhou et al both used inactivated or subunit vaccines and demonstrated MatAb interference (now lines 439 to 441). Yang et al did vaccinate pups with a very similar strain of murine rotavirus to Muleta et al, but whereas Yang et al administered 10x shedding dose 50%, Muleta used 1x10⁴ diarrhea dose 50%. Though it is not possible to directly compare these units, it is reasonable to

conclude that Muleta et al used a much higher dose of rotavirus, with shedding doses lower than diarrhea-inducing doses. We have added a comment about this in the discussion.

Line 466: I would use a different word than 'shedding' here. Shedding usually refers to viral RNA/particles/infectious virus - a more appropriate word for antibodies would be 'transfer'.

This is a good point! We've opted to use the word 'excretion' to reflect the fact we are specifically referring to antibodies in stool (now line 489). We have been through the text and replaced every mention of antibody shedding to antibody excretion.

Line 468: Can easily test this by measuring total IgA in stool (as mentioned above)

Yes, this has been addressed above

Line 493-494: Can you provide a citation for this?

We have edited our original statement 'by weaning at 3 weeks of age, comparable total MatAb transfer has occurred in mice relative to humans' to clarify that by focusing our study on seroconversion from 4 weeks of age (1 week post weaning), MatAb transfer will be complete irrespective of the route of delivery (now line 521 to 522).

Line 512: I would use a different word than blighted. Blighted seems too strong as despite the side effects, the world still has two effective rotavirus vaccines that are used in pediatric vaccination schedules.

We agree, and have changed this to 'challenged' (now line 543).

Line 576-577, 590: Could you please mention the source of your rotavirus-specific polyclonal antibodies (sheep and rabbit) in the manuscript?

These are both commercially available antibodies as listed in the Reagents and Tools Table.

Referee #3:

The primary aim of this study was to elucidate the mechanisms by which maternal antibodies (MatAbs) interfere with the efficacy of live-attenuated rotavirus vaccines in human infants. The authors successfully established and standardized a neonatal mouse model to mimic MatAb interference, representing a significant methodological advancement. This innovative model allowed the authors to explore the suppression of active IgG and IgA antibody responses in serum and feces and the reduction in B and T cell germinal center formation within the mesenteric lymph nodes (MLN). Additionally, the study demonstrated that MatAbs significantly impair vaccine live-attenuated virus replication in the gut. Importantly, the authors showed that FcγRIIB signaling was not essential for MatAb-mediated interference. The study is also the first to report single-cell analysis of the lymph node response to rotavirus infection, providing valuable insight into the anti-viral pathways stimulated.

Overall, the study provides compelling evidence that the primary mechanism of interference involves MatAb-mediated clearance of the vaccine virus before it can effectively engage the neonatal immune system. This work is a well-designed and carefully executed study that offers important insights into rotavirus immunology. Its findings are highly relevant not only to the field of rotavirus vaccination but also to our broader understanding of maternal antibody interference in oral vaccine responses.

Thank you for this overview

2- Specific major concerns essential to be addressed to support the conclusions

2.1- A major concern is the lack of discussion and consideration of other components of maternal immunity beyond antibody transfer. Recent studies have highlighted that, in addition to maternal antibodies, immune cells and cytokines are also transferred from mother to offspring - both transplacentally and via breastfeeding - and can play a significant role in shaping neonatal immune responses. This aspect is not addressed in your study. Please justify why these components were not included in your so deep analysis and discuss how their omission might affect the interpretation of your findings, particularly in the context of early-life vaccination strategies.

Thank you for highlighting this valuable issue. We fully acknowledge that maternal immunity encompasses more than antibody transfer alone, and that immune cells and cytokines transferred both transplacentally and via breastfeeding can influence neonatal immune development. Our study focused specifically on identifying how MatAb transfer functionally impacts early-life vaccine responses. This narrow scope was intentional and designed to isolate the role of MatAbs in modulating vaccine responses during a defined early-life window. We view this work as a necessary first step, upon which future studies can build to include additional maternal immune components. We have now added this limitation of our work in the discussion (now lines 528 to 532), explaining that whilst our findings contribute important insights into the antibody-mediated dimension of maternal immunity, they should be interpreted within the context of this more narrowly defined mechanism.

2.2- Another major concern is the absence of virus neutralization assays in your evaluation of the antibody response. While ELISA can provide information on the presence of antigen-specific binding antibodies (IgM, IgA and IgG), it does not indicate whether these antibodies are functionally capable of neutralizing the virus. Given that protection is more closely associated with neutralizing antibody titers, this omission limits the interpretation of vaccine efficacy.

Thank you for raising this issue about neutralization assays. To address this, we have performed neutralization assays on serum samples collected from mice vaccinated with the human rotavirus strain. Neutralization assays require strain-specific virus as extracellular neutralization is mediated by antibodies specific for VP4 and VP7, so we were readily able to perform human rotavirus-specific neutralization assays with these samples. The neutralization titers for these 18 mice are presented in a new figure 1G, which shows excellent concordance with the ELISA values. This provides evidence that results from our ELISA assays are in agreement with more functional assays. These methods have now been included in the Methods and Protocols section.

However, it is important to note that whilst we agree that for many virus infections, neutralization titers are important correlates of protection, for rotavirus infections there is clinical evidence this is not always the case. Clinical trials have shown limited correlation between neutralization titers and vaccine efficacy. An important example is provided in the recent PATH phase III clinical trial, whereby almost 100% infants made neutralizing antibodies to the subunit vaccine in the phase I/II trial (Groome et al 2017), but the subunit vaccine proved inferior to an existing live attenuated vaccine. Neutralizing antibody titers have been reported to 'consistently underestimate' the protection induced by rotavirus vaccination (Clark and Desselberger 2014).

The most definitive method for evaluating protection from viral infection in mice is to perform virus challenge experiments. In this study we used this approach to test the protective capacity of RV-specific antibodies in our mice, presented in Figures 5 and 8 (new data). Results in Figure 5H demonstrate that serum RV-specific IgG and stool IgA responses (Figure 3 E and F) are directly and inversely correlated with viral shedding.

Additionally, ELISA results are presented as optical density (OD) values at a single sample dilution, which can be misleading and may not reflect the actual magnitude of the response. It is standard practice to report ELISA titers as the reciprocal of the highest dilution yielding a positive signal. Please consider including these data or discussing these limitations more explicitly.

We agree that whilst serial dilutions of samples collected at each time point would be valuable for determining antibody titers, this was not technically possible - for longitudinal serum IgG sampling of the same mice we collected blood from the saphenous vein every two weeks starting from 4 weeks of age. The volume of blood obtainable from a 4 week old mouse only permits ELISA testing at a single dilution. For continuity and comparison across the weeks we chose to analyze all future timepoints at the same dilution. This has now been described in the results section text (now lines 109 to 111). The longitudinal curves we observed for IgG titers across multiple graphs shows that we have been able to capture dose-dependency using this approach.

Lines 38-39:

In the Introduction, it is important to explain the species-specific expression of the neonatal Fc receptor (FcRn) in the placenta. For example, horses and cattle do not express FcRn in the placenta, and therefore, there is no transplacental transfer of maternal antibodies in these species. In contrast, mice express FcRn both in the placenta and in the neonatal intestine, allowing maternal IgG transfer during gestation and also postnatally through colostrum and milk. Please clarify how this mechanism operates in your model vs humans. Is FcRn expressed in the human placenta and neonatal gut? Justify why the mouse is considered an appropriate model

for studying maternal antibody transfer in humans, and discuss the limitations of using this model and not a primate model.

We completely agree there are some key biological differences between MatAb transfer in different species, which are highly dependent on FcRn expression. We have added two sentences (now lines 78 to 81) to the introduction to highlight that IgG transfer in humans is predominantly transplacental, whereas in mice it is transplacental and via milk. This is then extrapolated in the discussion when considering limitations of our model (now line 517 to 521), citing Appleby and Catty's paper that detected 30% maternal IgG in mouse pups at birth, which is lower than the ~100% in human neonates. In the introduction we have explained that both the tractability of mouse models, and the fact that both mice and humans have hemochorial placentas are valuable strengths of using mice to study MatAb transfer.

Line 95:

What was the rationale for choosing day 7 postnatal as the time point for vaccination of pups? You mention this is meant to mimic human infant vaccination at 8 weeks of age - please explain how this timing aligns developmentally and immunologically between mice and humans.

This important question was also raised by reviewer one, and we have addressed this above and in the manuscript (now lines 100 to 103, 513 to 515).

More importantly, have you thought in testing for the presence of IgM, IgA and IgG antibody-secreting cells in the gut and in the spleen? You need to check for the activated cells that could be primed even in the presence of very high titers of maternal IgG. In calf models of RVA it was reported that IgM switching to IgA and mostly IgG1 was impaired by colostral passive antibodies, but most importantly the active immune response was modulated due to the cytokines of the colostrum not only the antibodies, what can you say about that in your model?

Thank you for this suggestion, this is very interesting to consider. We agree that focused evaluation of RV-specific antibody-secreting cells would be valuable for future studies. The data we have presented in this manuscript does provide preliminary evidence that it is possible for MatAbs to interfere with just one antibody isotype (Fig 4C, no serum or stool IgA was detected in FcyRIIB KO mice), which suggests potential involvement of MatAbs with class switching. We agree this will be an important avenue for future research.

It is very interesting to consider how non-antibody factors such as cytokines in milk may influence neonatal immunity, and we have now added consideration of this to the discussion (lines 528 to 530). However, we have still seen interference in our model after weaning (i.e. Figure 8A when pups were vaccinated at 4 weeks of age, but weaned at 3 weeks old), so this argues against non-antibody factors in milk solely mediating the interference we observed.

Lines 101-108:

You evaluated IgG and IgA antibody responses in serum and feces. However, no data are shown for IgM responses. Could you explain why IgM was not assessed, considering it is the first isotype produced during a primary immune response and could provide additional insights into vaccine immunogenicity and interference due to maternal antibodies

Thank you for this valuable suggestion, we had not previously examined IgM responses as IgM is not transferred placentally and only minimal amounts are transferred in milk. Therefore we did not expect to detect any IgM MatAbs in pups. However we agree that analysis of IgM in pups would be interesting to determine if MatAbs block antibody production at this very early stage of a primary antibody response, or whether interference occurs at the class switching level.

In light of this, we modified our rotavirus-specific sandwich ELISA to detect IgM, and detected a low level of RV-specific IgM on serum samples collected 7d post infection in adult mice. However, when the same methodology was applied to serum collected from naive mice 7 days post infection (terminal cull at 14 days old), we were unable to detect any antibody when serum was diluted 1:50. This likely reflects a detection limit for our assay, but also highlights the minimal contribution that IgM is making to the overall antibody pool.

As an alternative approach we also turned to our single cell results and interrogated these for activated IgM-producing B cells in vaccinated naïve pups, and vaccinated pups that received MatAbs. We searched for B cells expressing Ighm with evidence of activation e.g. CD83. However, no distinct population of IgM-producing

B cells emerged in either group, and we suspect this could be due to the 14 day timepoint studied. Class switching had already occurred as evidenced by increased Aicda expression in the no MatAb group. In future work we could explore an earlier timepoint to address this question in more detail, but ultimately our goal is to understand how to induce a robust class switched neonatal antibody response. We have added consideration of this valuable point to the discussion (now lines 471 to 474).

Lines 397-407

Another important point to address is the rationale behind the assumption that parenteral vaccination with a double-layered particle (DLP) vaccine would be more effective in overcoming the interference of maternal antibodies than oral administration of a live attenuated virus. Given that live oral vaccines can replicate at mucosal surfaces and potentially induce local immunity despite the presence of maternal antibodies, it is not immediately clear why the parenteral route using a non-replicating antigen would be expected to perform better in this context. Please justify this choice and clarify whether any comparative data or previous studies support this hypothesis.

We completely agree that a live oral vaccine would be expected to induce a better immune response than a parenteral route. However, several other studies have shown that parenteral routes can overcome rotavirus-specific MatAbs as we mentioned in the text (Yang et al 2019, Johansson et al 2008.) A parenteral non-replicating vaccine also has the advantages that it could be administered in combination with other injectable vaccines, and it can be given to immunocompromised individuals. Moreover, a parenteral non-replicating rotavirus vaccine was in recent phase III clinical trials for rotavirus, thus a better understanding of this type of vaccine in the face of MatAbs is important.

Furthermore, while you observed an increase in ELISA signal following parenteral DLP vaccination in the presence of maternal antibodies, it is important to consider that maternal serum IgG - which can be systemically absorbed in mice - may also neutralize the DLP vaccine, potentially impacting its effectiveness. An increase in ELISA-detected binding antibodies does not necessarily correlate with protective immunity. Did you assess seroconversion in terms of neutralizing antibody titers, which are more directly associated with protection? Including this information would greatly strengthen your conclusions regarding vaccine efficacy in the context of maternal antibody interference.

We agree that further characterization of our observations with DLP-based vaccination and MatAbs would be valuable and we were able to repeat this experiment in two new litters of pups to generate additional samples and extend our conclusions. The serum IgG results from these additional litters mirrored the first litter, and we have been able to add this data to figure 8D. However, neutralizing assays traditionally measure antibodies specific for the outer capsid proteins of rotavirus (VP4 / VP7), which are absent from DLPs. Standard neutralization assays are therefore not expected to be useful in this context. Instead, we were able to perform our intracellular neutralization assay (which we have previously used to quantify the functional activity of DLP-specific antibodies (Caddy et al 2020)) to evaluate vaccine effectiveness, with the results presented in figure 8F. This showed that serum from parenterally-vaccinated pups were able to neutralize virus more effectively than orally-vaccinated pups. In addition, we are pleased to be able to add to the manuscript new results of a challenge experiment with mice vaccinated with DLPs in the presence of MatAbs, to serve as the gold standard for understanding protective efficacy. New figure 8G shows that parenteral administration of DLPs induces greater protection in the presence of MatAbs than DLPs administered orally. This is in agreement with the ELISA and intracellular neutralization data.

3- Minor concerns that should be addressed

Section in line 543

In M&M please provide the virus titer in FFU/ml also and compared with the dose of the vaccines used in humans.

We have now included in the M&M that adult mice were infected with 10 FFU EMcN rotavirus (100 uL of 100 FFU/mL) to induce sero-conversion and 7-day old pups were vaccinated with 0.01 FFU EMcN (50 uL of 0.2 FFU/mL).

Host tropism makes a direct comparison between vaccine dose across species difficult. However, the pup vaccine dose that was used here was selected based on the ability to induce sero-conversion (detectable serum IgG and stool IgA) in >90% of naïve pups without inducing clinical disease (diarrhoea) in this mouse

model. We've further shown this vaccine dose is capable of significantly decreasing viral shedding in a challenge model by almost 3-fold. Vaccine trials in human infants similarly induced high rates of sero-conversion (over 90%) with high rates of efficacy (over 90% protection against severe disease).

It will be also useful to estimate the ug of DLP administered in the parenteral vaccine, as we know that at least $10E6-10E7$ TCID₅₀/dose are needed to see a satisfactory response with inactivated whole virus.

We vaccinated pups with 10ug of DLP (as described in the results section). This is in line with 5-50ug dose suggested in Cold Spring Harbour Protocols for immunizing animals.

Section in line 575. Why did you not test the Ab titer in serial four-fold dilutions?

We have explained our rationale for using single point dilutions above; we agree this would be ideal but were significantly limited by sample volume for longitudinal sampling.

Why IgM Abs are missing?

We have addressed this issue in our comments above. We agree analysis of IgM will be valuable for further clarification of the mechanisms of MatAb interference, but ultimately our end goal in this study was to understand how a protective class-switched antibody response is induced.

Section in line 607. Why you did not do ELISPOT of ASC in the cell obtained from MLN, or other immune tissues?

We used flow cytometry to quantify germinal center B cells in MLNs, which also provided us with data regarding activated B cells from different groups of mice. We agree ELISPOTs would provide rotavirus-specific B cell activation, but given the very clear differences between groups of mice following rotavirus vaccination this wasn't deemed necessary for this study.

In the figure 1. (F) and figure 3 (E) and figure 4 (C) it is stool IgG instead of serum IgG?

Thank you for checking, but we can verify that each of these figures are measuring serum IgG. There is minimal IgG detected in stool.

I think it is an excellent research, congratulation to the author for the hard work!

Thank you, we are very pleased to receive this positive feedback.

Dear Sarah,

Thank you again for submitting your revised manuscript (EMBOJ-2025-120856R) to The EMBO Journal for our consideration, and for your patience during peer review. Your manuscript has been sent back to the three original referees who had previously assessed the first version of your manuscript, and we have now received their comments, which you can find below.

I am very pleased to say that all three referees are satisfied with the revision, acknowledge that the initially raised concerns have been successfully addressed, and now support the publication of your manuscript in The EMBO Journal after a few minor corrections. Please see below a list of minor changes/corrections provided by referee #2 and address them in a final version of your manuscript, along with a point-by-point response detailing all changes to the manuscript.

From the editorial side, there are also a few changes we need you to make in the final version of your manuscript before we can proceed with its acceptance for publication:

- Thank you for providing the referees access to the deposited data at GEO. The reviewer access code can now be removed from the Data availability statement of the manuscript. Please make sure that all deposited data will be publicly available at the time of publication.
- Please provide the e-mail address of the corresponding author on the title page of the revised manuscript.
- As per our journal's policy, "data not shown" (on pages 14 and 15) is not permitted. All data referred to in the paper should be displayed in the main or Expanded View (EV) figures, or in the Appendix. Please add these data or change the text accordingly if these data are not central to the study and its conclusions, or properly cite the respective published sources if these data can be found elsewhere.
- We noticed that callouts for Fig. 7A are missing.
- "Supplemental Information" is not a correct callout - it needs to be updated.
- Please change "Materials and Methods" to "Methods".
- Please note that EMBO press papers are accompanied online by:
 - A) a short (2 sentences) summary of the findings and their significance,
 - B) 2-5 short bullet points highlighting the key results, and
 - C) a synopsis image in .jpg or .png format that is exactly 550 pixels wide and 300-600 pixels high (the height is variable). Please note that all text needs to be legible at the final size.Please upload this information along with your revised manuscript (the text for A and B should be provided in a separate Word file).
- We would be grateful if you could please re-organize your uploaded Source Data so that all Source Data for all panels of each main Figure are provided in a single zipped folder named "Figure # Source Data". Please make sure that no additional Excel files are provided outside of the zipped folders. For EV Figures, please zip all Source Data together in a single ZIP folder named "EV Figures Source Data".
- During our routine pre-acceptance Figure checks, we detected re-use of the same FACS dotplot between Figure 2D and Figure EV1E. Please check these Figures carefully and correct them if necessary, or -if the reuse is intentional and justified- please detail it in the respective Figure legends.
- During our routine data checks, our data editors have raised the following queries regarding figures, data, and legends. Please make sure that all requests below are completely addressed in the final version of your manuscript (please highlight all changes in the revised manuscript):
 - Please provide the exact p-values in the legends of Figures 1B-F; 2A; 3B, C, E, F; 4B, C; 5B, C, D, E, H, I; 8B, C, D, F, G.
 - Please indicate the statistical test used for data analysis in the legends of Figure EV5 H.
 - Please note that information related to "n" is missing in the legends of Figures 1C, D; 2A, D; 3D; 5D, E, H, I; 6G, 7D, 8C, D, F, G; EV2 A-C; EV3 A, B; EV4 A, B; EV5 H.
 - Please note that the error bars are not defined in the legends of Figures 1B-G; 2A, D; 4A-C; 8C-G; EV2 A-C; EV3 A-D.
 - Please note that the measure of center for the error bars needs to be defined in the legends of Figures 5A-I.

Please also note that as part of the EMBO publications' Transparent Editorial Process, The EMBO Journal publishes online a Peer Review File along with each accepted manuscript. This File will be published in conjunction with your paper and will include the referee reports, your point-by-point response and all pertinent correspondence relating to the manuscript. You can opt out of

this by letting the editorial office know (contact@embojournal.org). If you do opt out, the Peer Review File link will point to the following statement: "No Peer Review File is available with this article, as the authors have chosen not to make the review process public in this case."

We look forward to seeing a final version of your manuscript as soon as possible. Please let us know if you have any questions and use this link to submit your revision: <https://emboj.msubmit.net/cgi-bin/main.plex>.

Best regards,

Ioannis

Referee #1:

The authors were very thorough in their revision of the manuscript. All of our concerns have been sufficiently addressed.

Referee #2:

The authors have successfully addressed all my concerns. There remain only minor corrections:

-Line 68-70, suggestion to edit this sentence for a more optimistic spin for example 'Ultimately, each experimental vaccine model offers unique insights, and together they reveal diverse immune mechanisms...' or something similar. It's likely challenging to draw single conclusions because there isn't one conclusion for matAb interference but that's not a negative thing in this reviewer's opinion.

-Line 102-103, 375: Can you provide citation(s) that a 7-day old pup models a 6-8-week-old infant? Reviewer #1 challenged this assumption. It would be helpful to your argument if citations were provided.

-Line 103: Change 'genders' to 'sexes'

-Line 272: The majority of uncomplexed IgG is degraded in lysosomes if they do not bind FcRn. Phagocyte-mediated degradation is accelerated when IgG is complexed to an antigen. Please edit for clarity.

-Line 398-399: What do the authors propose is the reason why DLPs escape matAb interference?

-Line 458: '...rotavirus vaccination *at* MatAb titers...' it isn't clear what 'at' means in this sentence

This paper is robust and will significantly benefit the field of maternal/neonatal immunity and rotavirus vaccinology.

Referee #3:

The paper in the current form is ready for publication.

Our thanks again to the reviewers for their additional time reviewing the resubmission of our manuscript. We appreciate their consideration of our additional experiments and response to their reviews. We have addressed their final thoughts and comments below.

Referee #1:

The authors were very thorough in their revision of the manuscript. All of our concerns have been sufficiently addressed.

Thank you!

Referee #2:

The authors have successfully addressed all my concerns. There remain only minor corrections:

-Line 68-70, suggestion to edit this sentence for a more optimistic spin for example 'Ultimately, each experimental vaccine model offers unique insights, and together they reveal diverse immune mechanisms...' or something similar. It's likely challenging to draw single conclusions because there isn't one conclusion for matAb interference but that's not a negative thing in this reviewer's opinion.

Thank you for the suggestion. The sentence now reads: Ultimately, the diversity of immune responses needed to confer protection against distinct pathogens following vaccination may limit our ability to apply conclusions from one experimental vaccine model to another; however, each experimental vaccine model still offers unique insights and the ability to reveal diverse immune mechanisms. Now line 68 to 71.

-Line 102-103, 375: Can you provide citation(s) that a 7-day old pup models a 6-8-week-old infant? Reviewer #1 challenged this assumption. It would be helpful to your argument if citations were provided.

Thank you for the comment. As discussed with reviewer 1 and in the manuscript, we chose to vaccinate 7-day old pups to capitalize on the period when pups are readily susceptible to rotavirus and capable of mounting a rotavirus specific response. We believe this point is physiologically relevant to human infants during a similar period of immune maturation while infants may still be receiving maternal antibodies orally via breast milk and receive an oral rotavirus vaccination (6 to 8 weeks of age). Both neonatal pups in our model and humans infants at 6 to 8 weeks of age are able to mount a rotavirus-specific immunoglobulin response to vaccination. We have included a reference to human vaccination timing and the sentence now reads: 'This timepoint also approximately models the age at which human infants can mount an immunoglobulin response to rotavirus vaccination (6-8 weeks of age) (Armah et al., 2016)'. Now line 102 to 104.

-Line 103: Change 'genders' to 'sexes'

We have changed this to sexes. Now line 105.

-Line 272: The majority of uncomplexed IgG is degraded in lysosomes if they do not bind FcRn. Phagocyte-mediated degradation is accelerated when IgG is complexed to an antigen. Please edit for clarity.

This has been edited for clarity and now reads: 'Antibodies in complex with antigen in circulation are typically cleared by innate immune cells such as phagocytes or degraded in lysosomes if not bound by FcRn...' now lines 272 to 274.

-Line 398-399: What do the authors propose is the reason why DLPs escape matAb interference?

We agree that some DLP escaped MatAb interference to induce sero-conversion when given orally; however, only about half of pups sero-converted with or without maternal antibodies and therefore the response to oral DLP does not appear to be robust or dependent on maternal antibodies. We therefore used a parenteral approach to overcome maternal antibody interference and were able to improve sero-conversion rates to nearly 90% of pups. We propose that DLPs administered parenterally escape being cleared by MatAbs in the GI tract.

-Line 458: '...rotavirus vaccination *at* MatAb titers...' it isn't clear what 'at' means in this sentence

Apologies, we meant to indicate MatAb titers that have been induced by natural infection, the sentence has been changed to '...rotavirus vaccination *when* MatAb titers are induced by natural infection.'

This paper is robust and will significantly benefit the field of maternal/neonatal immunity and rotavirus vaccinology.

We agree!

Referee #3:

The paper in the current form is ready for publication.

Thank you for your help getting it ready for publication!

Dear Sarah,

Congratulations on an excellent manuscript! I am very pleased to inform you that it has been accepted for publication in The EMBO Journal. Thank you for comprehensively addressing the initially raised referee concerns and all editorial requests for corrections and changes.

There is only one minor change we need from you before we can move forward with the publication process: could you please remove from your synopsis image the line at the bottom regarding the graphics library that was used for the generation of the image, and instead add this information in the Acknowledgements section of the manuscript? Please send me (via e-mail) the revised synopsis image and the new Acknowledgements text, and I will add them to the files we will use for typesetting.

Your manuscript will then be processed for publication by EMBO Press. It will be copy edited and you will receive page proofs prior to publication. Please note that you will be contacted by Springer Nature Author Services to complete licensing and payment information.

If you have any questions, please do not hesitate to contact the Editorial Office. Thank you for your contribution to The EMBO Journal. Working with you has been a pleasure!

Best wishes,

Ioannis
